# Conformational landscapes of rigid and flexible molecules explored with variable temperature ion mobility-mass spectrometry

Xudong Wang[1], Emma Norgate[1], Junxiao Dai[1], Florian Benoit[1], Tony Bristow[2], Richard M. England ®[3], Jason M. D. Kalapothakis[1] & Perdita E. Barran ®[1] ✉

Understanding the effect of temperature to the structural integrity of proteins is relevant to diverse areas such as biotechnology and climate change. Variable temperature ion mobility-mass spectrometry (VT-IM-MS) can measure the effect of temperature on conformational landscapes. To delineate collision effects from structural change we report measurements using molecules with different degrees of rigidity namely: poly (L-lysine) (PLL) dendrimer, ubiquitin, β-casein and α-synuclein from 190-350 K. The CCS of PLL dendrimer varies with temperature consistent with collision theory, by contrast, the structure of each protein alters with notable restructuring at 350 K and 250 K, following predicted in vitro stability curves. At 210 K and 190 K we kinetically trap unfolding intermediates. For alpha-synuclein, the 13+ ions present two distinct conformers and VT-IM-MS measurements allow us to calculate the transition rate and activation energies of their conversion. These data exemplify the capacity of VT-IM-MS to provide insights on the thermodynamics of conformational restructuring.

Ion mobility spectrometry coupled with mass spectrometry (IM MS) has been widely used in the characterisation of molecular conformation. In one single experiment, it is possible to obtain the chemical identity and complex stoichiometry from $m/z$ values, and conformational information in the form of a collision cross section (CCS)[1–4]. Ions are created and injected into a cell which contains an inert buffer gas. These ions are propelled through the drift cell under the influence of a weak electric field and their arrival time at a detector is determined by their mass and charge as well as by the frequency and nature of the collisions that each ion makes with the gas. Mason and Schamp[5] related the average arrival time of ions to their rotationally averaged temperature-dependent CCS using a first approximation of the ion

transport equations. Consequently, the CCS is a composite parameter that contains information about the analyte ion, the buffer gas and the temperature of the gas. Commercially available IM-MS instruments do not allow direct measurements of the effect of temperature especially below 295 K on CCS values. The dependence has been noted and in pioneering experiments over 20 years ago, Bowers showed that that experimental CCS values of macromolecules were indeed temperature-dependent in mobility measurements[6,7], illustrating this with the use of $C_{60}^+$ and observing that as the temperature decreases the CCS increases[7]. This is attributable to the importance of the long-range interaction potential between the buffer gas and the analyte ion at low temperatures: in that case, the collision trajectory of the

[1]Michael Barber Centre for Collaborative Mass Spectrometry, Manchester Institute of Biotechnology, Department of Chemistry, The University of Manchester, 131 Princess Street, Manchester M1 7DN, UK. [2]Chemical Development, Pharmaceutical Technology and Development, Operations, AstraZeneca, Charter Way, Macclesfield SK102NA, UK. [3]Advanced Drug Delivery, Pharmaceutical Sciences, R&D, AstraZeneca, Macclesfield SK10 2NA, UK. ✉e-mail: perdita.barran@manchester.ac.uk

ion–neutral pair is influenced by long-range attractive forces giving rise to the larger CCS values observed. Conversely, at high temperatures the buffer gas analyte interactions become more hard-sphere-like, and the CCS approaches a value that is influenced to a lesser extent by interatomic interactions. Such behaviour is evident from the collision theory developed by Mason, McDaniel, Schamp, and Viehland[5,8], in turn informed with experiments on atomic and diatomic ions (Eq. (1)).[5] This behaviour can be predicted for rigid systems. When keeping the geometry of a molecule constant, the measured collision cross-section is inversely proportional to the square root of the temperature[5]:

$$\bar{v}_{\mathrm{d}} = \frac{3}{16} \frac{qE}{N} \sqrt{\frac{2\pi}{\mu k_{\mathrm{B}} T}} \frac{1}{\Omega} \qquad (1)$$

In Eq. (1), $N$ represents the number density of the drift gas, $q$ denotes the charge of the ion, $E$ is electric field, $k_{\mathrm{B}}$ is the Boltzmann constant, $\mu$ is the reduced mass of the ion of interest and the buffer gas; $T$ is the gas temperature and $\bar{v}_{\mathrm{d}}$ is the average drift velocity[9]. $\Omega$ is the momentum transfer integral of all ion-buffer gas collisions, also known as the collision cross section (CCS). Each term in this equation plays a pivotal role in determining the CCS, allowing for a precise and comprehensive understanding of ion interactions within the drift gas environment[10] at field strengths where the drift velocity varies linearly with the electric field. Under conditions of stable mass, charge, electric field, drift gas density, the effect of temperature is seen in this equation in the temperature ($T$) term as well as in the collision cross section ($\Omega$). As a further point of clarification, Eq. (1) contains the effect of temperature on the average momentum $\langle p \rangle$ of ion–gas pairs, which is conserved when collisions are elastic since, $\frac{\langle p \rangle^2}{2m} = \frac{3}{2} k_{\mathrm{B}} T$. Further temperature effects are included in the $\Omega$ term, the orientationally and conformationally averaged collision cross section. These can be classified as being of two kinds: first, the temperature dependence of the buffer gas–ion interaction potential, and secondly, for polyatomic molecules the effect of temperature on the structure of the ions.

On a conceptual level, the assumptions of Eq. (1) are that collisions between a single gas molecule and a single ion comprise the vast majority of all collision events, so that many-body effects can be effectively ignored, and that the collisions are elastic. Although a hard-sphere potential without other internal degrees of freedom would result in elastic collisions, the same would be true with potentials that become flat at long distances—i.e. at sufficiently large ion–neutral distances there will be conservation of kinetic energy. In summary, in order to derive $\Omega$ from a drift tube experiment (which is the drift velocity at a given $E$) it is important to include the temperature $T$ at which the collisions occur. The effect of that temperature on a given ion, which is explored here, is found by comparing the CCS obtained at each temperature.

Proteins are not rigid systems both in vitro and in vivo as functional forms of proteins are influenced by a variety of factors[11], including temperature[12]. For eukaryotic proteins, their fold follows a stability curve being most stable at physiologically relevant conditions and deviations from those to lower or higher temperatures inducing denaturing and the loss of the functional form[9]. Performing experimental measurements that explore the effect of elevated temperature on protein structure is straightforward with a number of different methods being available, including isothermal calorimetry[13], NMR[14] and temperature jump spectroscopy[15] as well as mass spectrometry based proteomic methods such as LOPIT[16]. These have led to a body of work that agrees that increased thermal energy overcomes the energy required for non-covalent interactions to hold a protein in its folded state[10,17]. Such measurements have also been used to support two state unfolding models for structured proteins and the absence of such for intrinsically disordered proteins (IDPs). By contrast, exploring

the effect of low temperatures on protein structure on globular proteins and IDPs is nontrivial and much of the work in this area is theoretical, based on extrapolation from measurements made close to 273 K[18] or with protic organic solvents which freeze at lower temperatures. The prevailing view from these studies is that as the aqueous environment around a protein freezes, it becomes more hydrophobic, which reduces the often-small stabilising influence of solvent interactions for the fold and causes the protein to denature[19].

It is well known that collisions in the gas phase can be used to alter the structure of proteins to gain insights to their topology and interactions, and IM-MS has been employed to examine unfolding and restructuring as well as dissociation. It is possible to estimate the temperature of the ion due to collisional interactions. VT IM-MS measurements also allow us to probe protein unfolding as well as monitor the effect of elevated temperature on conformational dynamics over 2–50 ms. Temperature effects in solution can also be measured by changing the temperature of the ESI solution and use IM-MS to examine how conformations are altered[20,21].

The effect of low temperature on protein conformations in the gas phase has been less studied, in part due to the instrument design challenges to include cooling, but it is a relatively unexplored environment for study. Russell and co-workers used cryogenic IM-MS to examine the conformational changes that occur during the hydration of a small peptide, one water molecule at a time[22]. By varying the time ions spend in the drift cell, it is also possible to make thermodynamic measurements, and monitor the rate of conformer interconversion, which has been shown for peptides and nucleotides[23]. A typical VT-IM-MS workflow is shown in Fig. 1, combined to a quadrupole time-of-flight instrument for $m/z$ separation. Changes to the temperature of the analyte can be made at different stages of the instrument. These include altering the temperature of the ESI source and hence the analyte solution[20] and increasing the voltages in the desolvation optics to induce in source collisional activation[24]. More commonly, activation is performed on desolvated ions prior to the drift region in ion transfer optics or collision cells. In many IM-MS instruments it is also possible to increase the energy at which ions are injected into the drift cell, and finally some instruments permit the use of cooled drift gases down to cryogenic temperatures[25,26].

Using VT IM-MS coupled with native IM-MS methods where a series of model structured proteins are infused from a salty solution that preserves the solution fold; intriguingly, we found that an increase in the CCS at 260 K indicates that the proteins will denature at low temperatures in the absence of solvent[27]. Using a higher resolution IM-MS instrument, we found that ubiquitin does not behave in this way, but rather more like a rigid system, however, following in source activation prior to injection to the drift cell we could capture unfolded intermediates with the use of low drift gas temperatures[26]. We have explored this effect further with monoclonal antibodies[28], where the role of low temperature has implications for the storage of biopharmaceuticals. Again, we found that these proteins extend at 260 K and that this restructuring was more pronounced for mAbs where the hinge was stiffer (IgG2 vs. IgG4)[29]. This remarkable finding caused us to consider what happened to inter or intra molecular interactions as a function of changes in temperature. We postulate that the hydrogen bonds involved in non-covalent interactions in the native structures lengthen and weaken as the protein cools, the residual energy at 250–275 K is sufficient to overcome the stabilisation they provide permitting restructuring.

IM-MS is highly suited to study conformational populations of intrinsically disordered proteins, that commonly present to the mass spectrometer with wide CCS and wide charge state distributions compared to more structured proteins[30,31]. Here, we examine the relevance of VT IM-MS applied to study conformationally dynamic proteins using, as exemplar systems, denatured ubiquitin, β-casein and

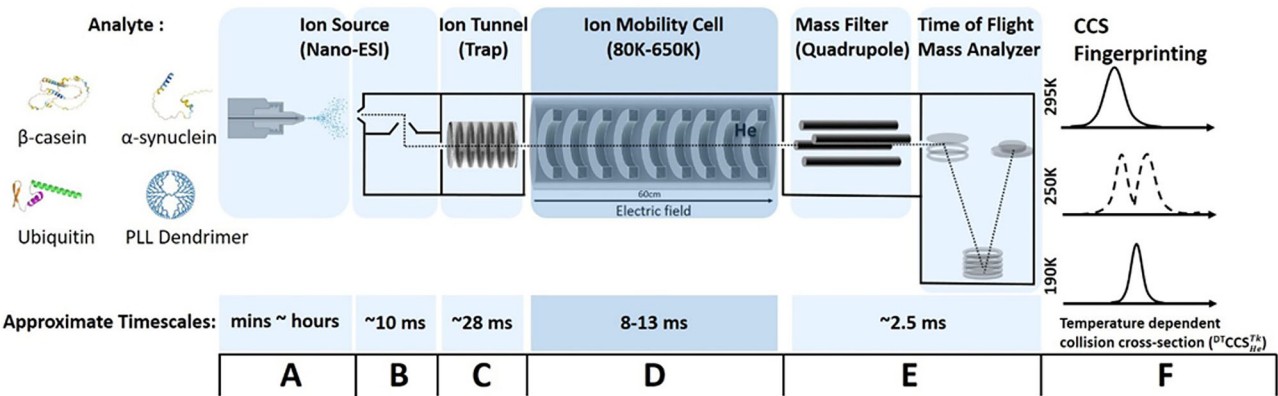

**Fig. 1 | Schematic workflow of VT-IM-MS experiments and the approximate time spent by ions in each stage in our instrument for the four analytes under investigation.** The structures of β-casein and α-synuclein are represented using AlphaFold[82], the 'A' state of ubiquitin is adapted from Brutscher et al.[41,83] and the PLL dendrimer is a schematic representation of the structure of a polylysine rich polymer as reported previously[55]. **A** A solution containing the sample of interest is placed in a nano-spray capillary, which can be heated at a given temperature for mins to hours[84]. **B** The time for an analyte to desolvate from the ESI droplet is 0.1–10 ms[85]. **C** Applying higher voltages in the source and transfer ion optics permits ion activation; ions are trapped for ~28 ms and released to ion mobility cell of length 50.5 cm in a 100 μs packet using gating voltages. In our instrument the temperature of ions in the trap is ~295 K even when the drift $T$ is 150 K[57]. **D** The drift time of ions is dependent on their mobility and the drift voltage and in this instrument for the proteins of interest ranges between 8 and 13 ms. Using cooling gases or heaters the temperature of gas in the drift cell can be altered from cryogenic to elevated temperatures and monitored with platinum resistance thermocouples in the cell. **E** Ions are transferred via a quadrupole ion filter into a time of flight (ToF) mass analyser all at ~295 K. **F** Ions are detected in the form of mass resolved arrival time distributions, which are converted to temperature dependent collision cross-section ($^{DT}CCS^{TK}_{He}$). This instrument can be used to examine the effect of temperature on an ion's mobility as well as on its structure. Approximate timescales based on values that are typical in the instrument used here for ubiquitin.

α-synuclein (Figs. 1, 2). We contrast their behaviours with that shown by a model polymer to allow us to decouple the effects of temperature on protein structure from the nature of the measurement. The structure of ubiquitin has been well studied with IM-MS[32–36]; and we have shown that for its low native like charge states, its CCS increases with decreasing temperature in a similar way to $C_{60}$[7]. By contrast, β-casein and α-synuclein are highly disordered with most of their sequences lacking stable secondary or tertiary structure (Fig. 2e)[37,38]. Whilst we selected these compounds as demonstrators for the benefit of using VT IM-MS each has an important biological or biotechnological role. The G5 dendrimer is an example of a class of molecule that has been developed for drug delivery with high biocompatibility[39]; ubiquitin is a small regulatory protein found in most tissues of eukaryotic organisms[40], its structure has been well studied by various techniques such as NMR and IM-MS[35,41]; β-casein is a major protein in milk that plays a crucial role in nutrient delivery and has high levels of disorder and conformational variability in lipid environments[42]; α-synuclein is a protein primarily involved in the regulation of synaptic vesicle trafficking and neurotransmitter release in the brain, it is implicated in the progression of Parkinson's disease[43]. For the proteins examined each has a different arrangement of charged residues which are differently segregated and for β-casein and α-synuclein (Fig. 2b), as for other IDPs this is a signature of their high disorder[44–46] which can be distinguished readily with IM-MS. We chose the G5 poly (L-lysine) (PLL) dendrimer (Fig. 2a) to compare the behaviour of these proteins to, since we have previously shown this family of dendritic polymers, possess stable and rigid conformations and G5 is a similar mass (8.02 kDa.) to ubiquitin (8.58 kDa.)

## Results

### Effect of varying the drift gas temperature on the CCS distributions

The IM-MS measurements for each of the analytes investigated show how this method can provide insights to their conformational preferences at different temperatures that for $T < 273$ K could not be obtained with any other existing analytical method (Fig. 3). For all species, at all charge states measured, the average $^{DT}CCS^{TK}_{He}$ value at

$T = 190$ are larger than at 295 K as predicted by the kinetic theory of ref. 47 gases[23], in line with what we have previously reported[26,29]. In order to study the effect of temperature in restructuring proteins (beyond what occurs to the collision cross section), we examine the net charge state just above that which could be supported on the surface of a globular protein[48]. These charge states are commonly more prone to unfolding/structuring from a globular form or native fold[48].

The temperature dependence in the arrival time distributions and collisional cross section distributions for each species is highly system dependent and, moreover, as their conformational diversity increases, G5 < ubiquitin < β-casein < α-synuclein, the measured CCS distributions also become more complex. We observe trends in behaviour that locate each analyte on a continuum between intrinsically rigid and highly flexible. The G5-dendrimer (Fig. 3a) behaves in a very similar fashion to that which has been previously reported for $C_{60}$, and in our prior work for native (low) charge states of ubiquitin, namely decreasing with temperature in a manner that is inversely proportional to the square root of the temperature in accordance with collision theory and increased influence of the long range attractive potential[7]. The $^{DT}CCS^{TK}_{He}/Å^2$ for $[M + 8H]^{8+}$ increases by ~1.3% from 295 to 190 K and slightly decreases from 295 to 350 K. The mode $^{DT}CCS^{TK}_{He}$ of other charge states exhibit the same tendency as well (Supplementary Table 1).

The $[M + 10H]^{10+}$ ion of ubiquitin (Fig. 3b), behaves as predicted by Gabelica and Marklund for calculations performed using the A-state of Ubiquitin[49], with a steady increase in CCS as the drift gas temperature decreases suggesting only small structural perturbation, although at 350 K there is evidence for some elongation suggesting the start of thermally activated unfolding[26,50]. Similar effects are observed for the other charge states of denatured ubiquitin (Supplementary Table 2) as the drift gas temperature is lowered the CCS increases by 1.33% for $[M + 8H]^{8+}$ and 4.1% for $[M + 11H]^{11+}$. In sub-ambient experiments, the mode $^{DT}CCS^{TK}_{He}$ of denatured ubiquitin in the higher charge state ($z = 7$–10) gradually increases with decreasing temperature as predicted according to Eq. (1)[5,49,51,52] (Supplementary Table 2). For $[M + 10H]^{10+}$ ions the CCS is 1912 Å$^2$ at 295 K and rises to 2198 Å$^2$ at

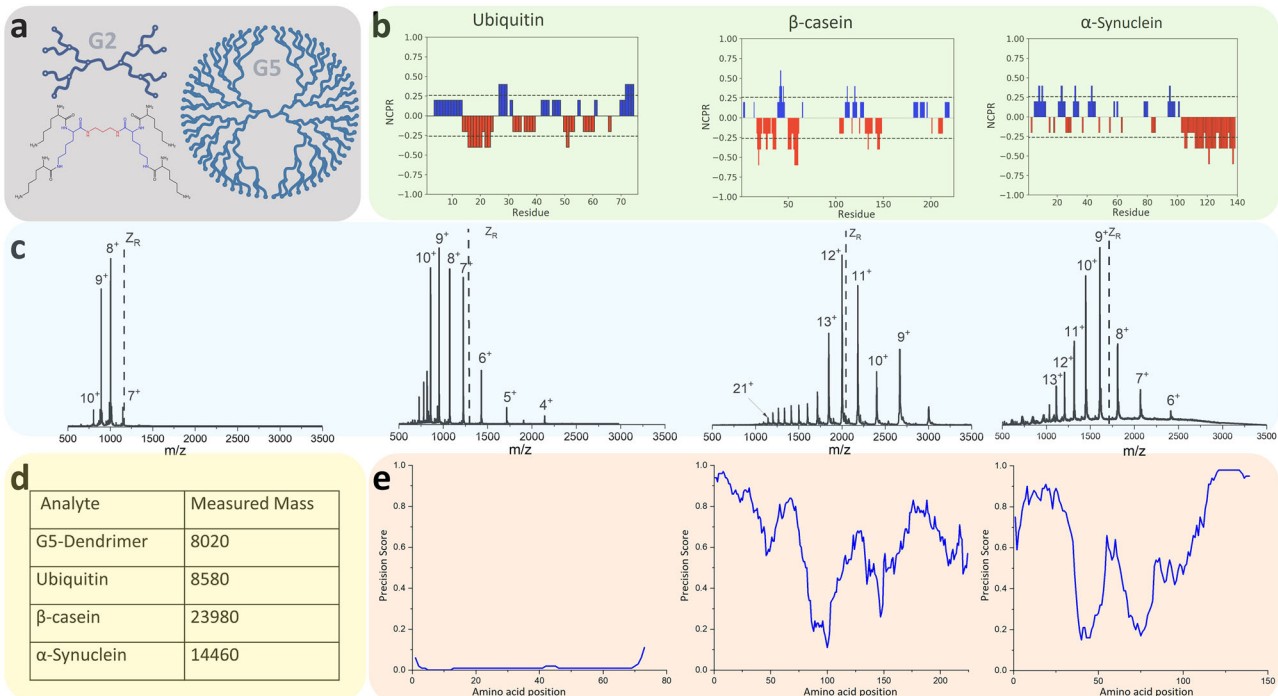

**Fig. 2 | Biophysical information regarding the analytes under investigation including net charge per residue profiles, nESI mass spectrometry data, the measured masses and the predicted disorder of the proteins under investigation[86]. a** Chemical structure of the G2 PLL dendrimer with the core (red), as well as the constituents of G1 (blue) and G2 (black) and schematic structures of G2 and G5[55]. **b** Net charge per residue (NCPR) profiles of construct along the linear sequence, adapted by CIDER[87]. **c** Typical mass spectrum of G5 PLL dendrimers from water, denatured ubiquitin sprayed from water: MeOH = 1:1, PH-2.0; α-Synuclein and β-casein sprayed from 50 mM ammonium acetate at T = 295 K, 20 μM, on the VT-IMMS instrument. The black dashed line $Z_R$ is the solution of an empirical relationship derived by De la Mora over the mass range of our test set of proteins, which describes the upper limit of charge that can be accommodated on the surface of a protein of a given mass if it remains quasi spherical[48]. **d** Measured mass of analytes. **e** DISOPRED plot for ubiquitin, α-Synuclein and β-casein.

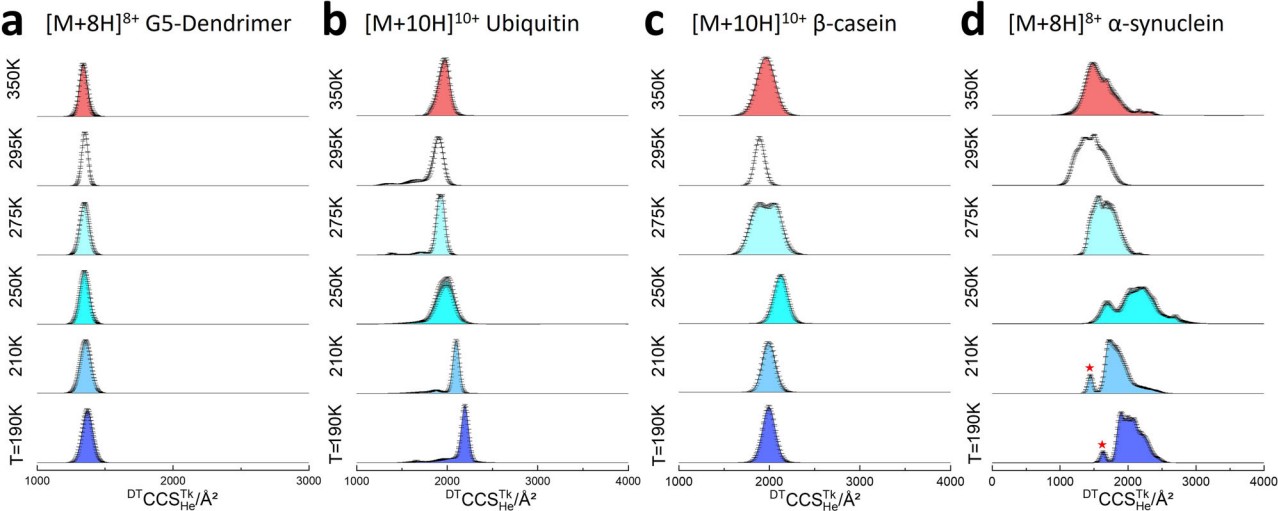

**Fig. 3 | CCS distributions for the most representative ions of the four systems studied.** $^{DT}CCS^{TK}_{He}$ data is shown for measurements made at T = 295, 275, 250, 210, and 190 K, respectively, with colour shading to denote each temperature for: **a** [M + 8H]8+ 5th generation PLL Dendrimer, **b** [M + 10H]$^{10+}$ ubiquitin, **c** [M + 10H]$^{10+}$ β-casein, and **d** [M + 8H]$^{8+}$ α-synuclein. Error bars represent the standard deviation from the average of three replicates−which is the measure of centre. We tried to maintain the E/N values within reason, and practically this was -10 Td. for 350 K and -5 Td for 190 K. Here is an interactive plot for data above.

190 K, corresponding to a relative increase of by 14.5%, which is above that predicted over this temperature range. For [M + 10H]$^{10+}$ there are also some smaller lower populated conformers, which appear to be better resolved at lower temperatures as is the main species. At room temperature the CCS values of these are 1355 and 1676 Å$^2$ which correspond to intermediates in the interconversion between the N and A state as described previously[35,41]. The benefit of low temperature IM

measurements is that it enables us to freeze out these species. The CCS of any gas-phase ion depends on both the effective temperature of the ion and the ion-neutral interaction potential. At lower temperatures, the long-range attraction between the analyte ions and the buffer gas is more dominant, whereas at higher temperatures, the interactions become hard sphere like. This is clearly exemplified by the behaviour of the G5 dendrimer, which exhibits a small decrease in CCS

when the DT temperature is increased to 350 K, in line with kinetic theory[5,7,53].

Given that the G5-dendrimer exhibits such ideal behaviour as the temperature is varied, it is useful to contrast it with measurements for proteins, especially those that are more conformationally dynamic as it allows us to distinguish the physics of buffer gas ion interactions, from the effect of the temperature of the buffer gas on the structure of the molecule.

The disordered proteins β-Casein (Fig. 3c) and α-synuclein (Fig. 3d) present a different conformational landscape at every temperature. At 350 K, both appear to start to thermally denature, as for $[M+10H]^{10+}$ ubiquitin. Unlike the G5-dendrimer, the $^{DT}CCS^{350K}_{He}$ of ubiquitin $[M+10H]^{10+}$ increases 5% at 350 K, and similar behaviour is apparent for β-casein and α-synuclein (Figs. 3-5). As the drift temperature decreases, for $[M+10H]^{10+}$ β-Casein primarily retains its unimodal CCS distribution, except at 275 K (see below). Other charge states show more complex restructuring (Fig. 4 and Supplementary Table 3). The behaviour at 210 and 190 K is as predicted, although the increase in CCS compared to the value at 295 K varies. For example, with $[M+10H]^{10+}$ it is 5%, whereas for $[M+16H]^{16+}$ it is 8.9%. With the high charge states of β-casein ($z > 20$) the net change in CCS between 295 and 190 K is only -1.5% which suggests that, like the dendrimer, these highly extended conformers are rigid forms that are primarily only influenced by the difference in buffer gas ion interactions at the lower temperature. α-synuclein shows far more complex behaviour with the change in drift gas temperature, as previously reported[26]. For the $[M+8H]^{8+}$ ion, at temperatures below 250 K, a very distinct conformer with a CCS of 1445 Å² at 210 K and 1632 Å² at 190 K is revealed, which is not resolved at higher temperatures suggesting that it is preserved by cryo-freezing when injected into the cold drift cell, and that either it restructures to a more extended form at temperatures 250 K and above or it is not able to be resolved from the myriad of other conformers with higher CCS values for this charge state. If we compare the predicted maximum resolution $R_{Max}$ for the dendrimer with the value from experiment we find that the resolution does not increase as we lower the temperature (Supplementary Table 4). This apparent discrepancy for α-synuclein is likely due to a number of closely related conformers that anneal at room temperature and start to separate at lower $T$ (Supplementary Tables 4 and 5). This said, the sharp peaks observed for both ubiquitin and α-synuclein are approaching $R_{Max}$ at lower $T$. In order to rule out the appearance of conformers due to charge reduction we perform activated ion mobility spectrometry (aIMS) experiments wherein we $m/z$ select each ion prior to activation and in doing this we do not observe any charge-reduced ions at low collision voltages. We have previously demonstrated this effect for ubiquitin[54] and G5-dendrimers[55]. The mass spectra obtained following activation for $m/z$ selected α-synuclein and β-casein are reported in Supporting Information (Supplementary Fig. 6; Fig. 8). Thus, we infer from low temperature IM-MS and aIMS experiments that charge stripping does not contribute significantly to the low temperature cases that we examine with linear-field IMS; further, there is no argon or nitrogen gas in our home-built instrument post drift cell (we do not fill the collision cell), unlike in the Synapt or other commercial IM-MS devices, which also diminishes the opportunities to charge strip. Another explanation for the myriads of conformers might be solvent adducts that are better retained at lower temperatures. We do not observe any significant intensity from water–protein ion clusters in our measurements at any temperature (Supplementary Fig. 10). We have previously examined the effect of salt adduction as a function of lowered temperature, with small structured proteins and found it can be higher at lower temperatures for some proteins, but we used high concentrations of NaI (2 mM) to detect protein-salt adducts[27]. In that work, which inspired this study, we saw only a marginal change in CCS due to salt adduction. Herein, we present CCS distributions from post-acquisition $m/z$ selection of ions that are both solvent-free and salt-free when detected although if we examine the ATDs of the weak peaks due to sodiated forms we find that they are highly similar to those of the bare protein ion (Supplementary Fig. 10).

In summary, the IM measurements at temperatures below 250 K show higher resolution than at room temperature, which exemplifies how such measurements may be useful to resolve low energy conformers. Many living organisms on earth need to survive temperatures between 250 and 350 K. Whilst it is possible with other biophysical method to explore the effects of temperatures at and above 275 K the region from 250 to 275 K is far less tractable, our data highlights how IM-MS is able to probe this temperature range.

## Effect of drift gas temperatures from 250 to 275 K on analyte structure

The conformational behaviour of each of the proteins studied at 250 and 275 K can not be attributed solely to the increased resolution at lower drift gas temperatures due to slower diffusion, nor the effects of buffer gas - ion interaction (Fig. 3). For Ubiquitin at 250 K, the measured $^{DT}CCSD^{250K}_{He}$ is significantly wider, indicating that some or all of the conformers present at room temperature restructure to larger forms. Such effects were not observed for the lower charge states of ubiquitin associated with the highly stable native fold[26], although we have previously reported similar restructuring behaviour for native charge states of intact and disulphide-reduced lysozyme, bovine pancreatic trypsin inhibitor (BPTI) and myoglobin[26,27].

For IDPs, this temperature-dependent change is also observed albeit with more complicated CCS distributions: taking $[M+10]^{10+}$ of β-casein (Fig. 3c) as an example, the mode of the $^{DT}CCSD^{295K}_{He}$ measured at 295 K in this experiment is 1894 Å², consistent with previous published data[56]. When the temperature drops to 275 K, the $^{DT}CCSD^{275K}_{He}$ now comprises of two species of similar intensity: one (1938 Å²) which is 2% greater than the value at 295 K and a second conformer at ~2000 Å². We infer from this result that approximately half of the population of ions present now have restructured to a more extended conformer. At 250 K, the $^{DT}CCSD^{250K}_{He}$ of casein is again unimodal, and now centred on 2050 Å², an increase of 8% compared to $^{DT}CCS^{295K}_{He}$, an increase far greater than that shown by the G5 dendrimer, implying that at this temperature all casein conformers elongate. We interpret this behaviour as cold denaturation of the protein in a gaseous environment. The other dominant charge states of β-casein behave in a similar fashion (Fig. 4) although the magnitude of the increase at 250 K is lesser for higher charge states.

For β-casein, the increase in the average $^{DT}CCS^{250K}_{He}$ of $[M+9H]^{9+}$, $[M+10H]^{10+}$, and $[M+11H]^{11+}$ are 10.7%, 8.2%, and 7.7% whereas for $^{DT}CCS^{250K}_{He}$ of $[M+21H]^{21+}$ and $[M+22H]^{22+}$ the increase is only 2.1% and 4.0%. We speculate that this may be related the number of stabilising non-covalent interactions that an ion at a given charge state has as it enters the cold drift cell. For low charge states we have previously postulated that the H-bonds, which stabilise more compact conformers, extend as the temperature of the ion decreases. At temperatures between 250 and 275 K there is sufficient residual energy in the protein that this lengthening of H-bonds leads to the decoupling of regions that are connected with these H-bonds. The higher the charge state, the fewer of such interactions exist, as they have already been reduced by coulombic repulsion due to proximal protons, and the less perturbation to these can occur. To investigate this behaviour further, we performed experiments which couple collision induced activation with IM measurements (aIMS) on β-casein, (Supplementary Fig. 5) which show similar effects. The lower charge states restructure via stable intermediates, presumably due to stabilising non-covalent interactions, requiring more collisional energy to fully extend, whereas higher charge states restructure with far less energy and via less stable intermediates, indicating that coulombic effects are the main driver.

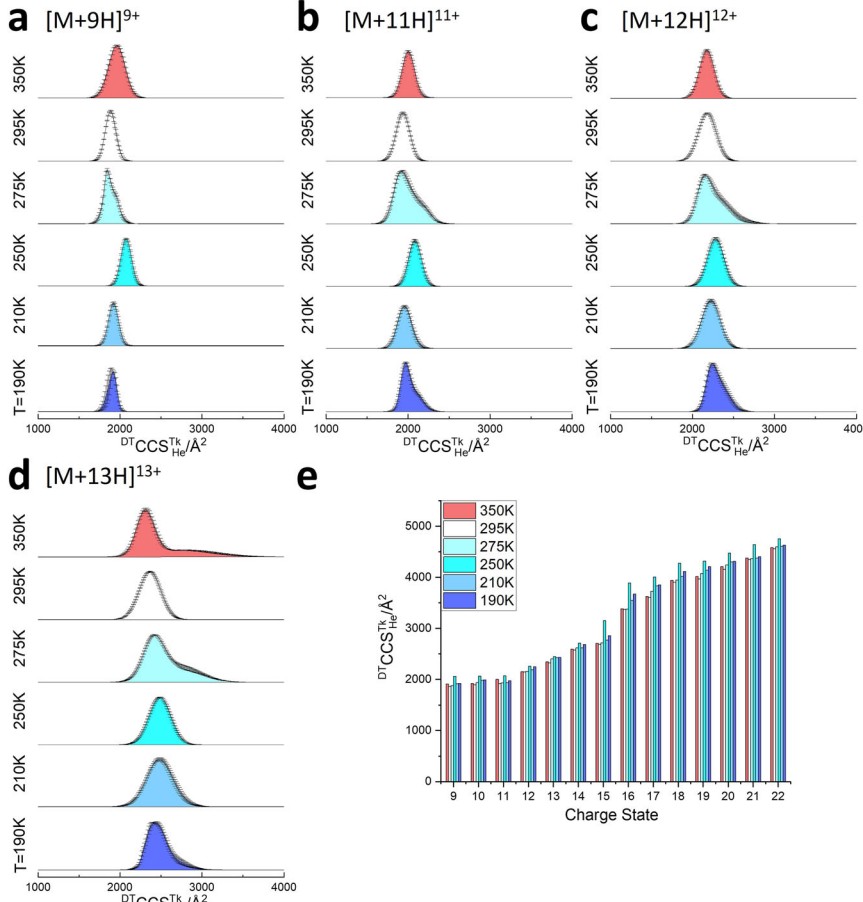

**Fig. 4 | β-casein collision cross section distributions ($^{DT}CCS^{TK}_{He}$) for the four most abundant ions. a** $[M+9H]^{9+}$, **b** $[M+11H]^{11+}$, **c** $[M+12H]^{12+}$ and **d** $[M+13H]^{13+}$ in the temperature range of 295 K to 190 K. Error bars represent the standard deviation from three replicates. Sub-ambient temperature median $^{DT}CCS^{TK}_{He}$ values for all other detected charge ($[M+9H]^{9+}$ - $[M+22H]^{22+}$) states are shown in (**e**). Here is the interactive plot for data above.

NCPR and disorder predictions (Fig. 2b, e) indicate that α-synuclein may have relatively higher conformational diversity than β-casein, since the opportunity to form many different stabilising interactions is provided by segregated patches of oppositely charged amino acids. This hypothesis is supported as well by ambient temperature experiments where the CCS distributions for the low charge states $[M+7H]^{7+}$ and $[M+8H]^{8+}$ are broad (Supplementary Fig. 4). For higher charge states ranging from $[M+10H]^{10+}$ to $[M+13H]^{13+}$, two or three distinct conformations are observed at room temperature. In this study these are better resolved than in our previous exploratory reports[26,29,57] and the data here is highly reproducible which we attribute to more careful sample preparation which reduced the contribution from counter ions on the protein as well as lowered the potential for conformers that are dissociated from higher order aggregates.

For all charge states of α-synuclein, significant restructuring occurs at $T = 250$ K. For the intermediate charge states $z = 9, 10$ and $11$, this is also evident at T = 275 K. For $[M+8H]^{8+}$ at $T = 250$, the average $^{DT}CCS^{250K}_{He}$ increases by 44% from 1510 Å² at 295 K to 2180 Å², *cf.* β-casein where, as discussed above, the increase ranges from 2% and 17% (Supplementary Table 3). This behaviour contrasts with $[M+7H]^{7+}$, which behaves more akin to low charge states of ubiquitin, with less evidence of global restructuring and CCS distributions that alter mostly due to ion mobility effects, indicating that the conformers of $[M+7H]^{7+}$ contain many stabilising non-covalent interactions. At 275 K for $[M+11H]^{11+}$ and higher charge states, the $^{DT}CCS^{275K}_{He}$ are better resolved, with the apex $^{DT}CCS^{275K}_{He}$ increased more than for the

dendrimer at this temperature, indicating restructuring to more distinct conformers.

For $[M+7H]^{7+}$ and $[M+8H]^{8+}$, such changes at $T = 250$ K are far less significant, but still evident. For the most abundant intermediate charge states (9+ and 10+) at $T = 250$ K, there are now conformers which have a median $^{DT}CCS^{250K}_{He}$ increase of 33% and 25%, respectively. Whilst this behaviour is similar to that shown by β-casein, the relative magnitude of the restructuring to larger conformers is greater.

For α-synuclein we carried out additional measurements at 225 K to determine if the conformational changes seen at 250 K only happen at that temperature (Fig. 5 and Supplementary Fig. 3). For the high charge state ions ($z = 10$–$13$) the $^{DT}CCS^{225K}_{He}$ distributions are very similar to those at 275 K with the slight increase in CCS magnitude expected at this temperature. The width of the conformers at 225 K is surprisingly wider than at 275 K and there is some evidence for conformers that are not resolved at higher temperatures, starting to be distinguished, most evident for $z = 10$ and $11$. Ions with $z < 10$ also exhibit a larger than expected increase in CCS at 225 K. The same effect is most pronounced for $[M+9H]^{9+}$ and $[M+10]^{10+}$, which possess very similar $^{DT}CCS^{225K}_{He}$ distributions as for $^{DT}CCS^{275K}_{He}$. This suggests a wider temperature range over which the low temperature denaturation can occur to initially compact ions.

In the lowest temperature experiments (210 K and 190 K), $[M+7H]^{7+}$ exhibited similar features as those of β-casein and other structurally rigid analytes: the median $^{DT}CCS^{TK}_{He}$ was slightly higher than at ambient temperature and the $^{DT}CCS^{TK}_{He}$ range was slightly narrowed. But for $[M+8H]^{8+}$ (Fig. 3d), we captured one additional

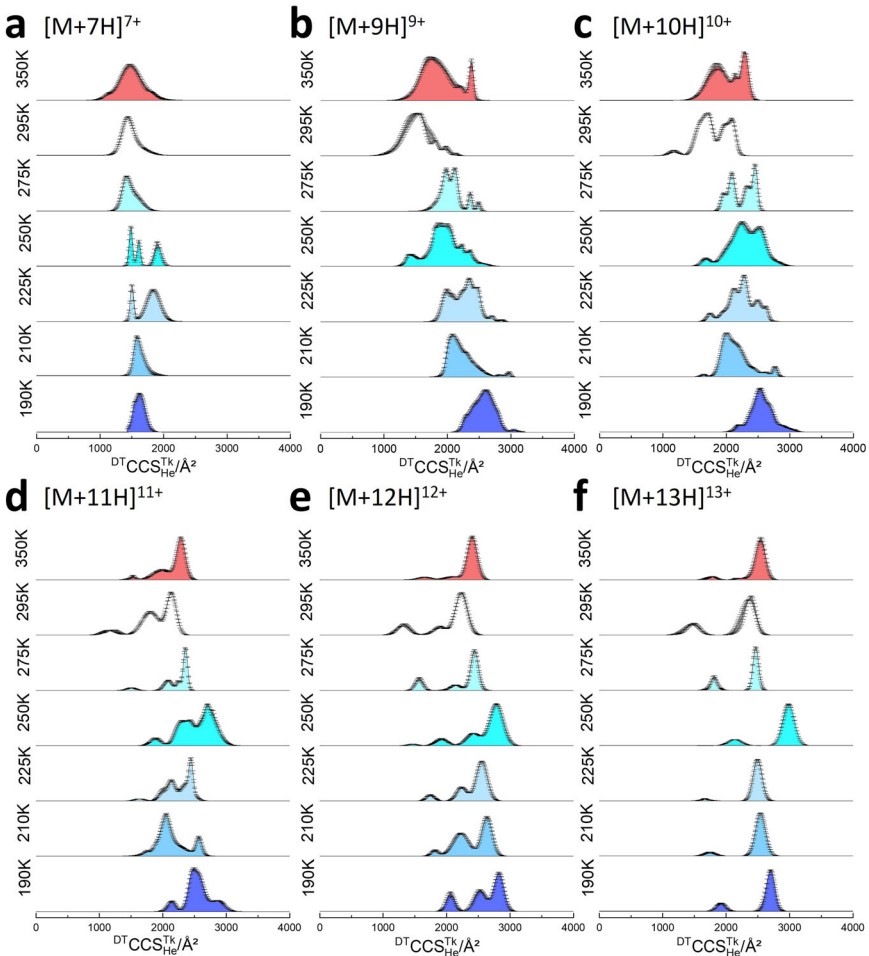

**Fig. 5 | α-synuclein collision cross section distributions.** $^{DT}CCS^{TK}_{He}$ for selected ions across the charge state distribution. **a** [M + 7H]$^{7+}$, **b** [M + 8H]$^{8+}$, **c** [M + 9H]$^{9+}$, **d** [M + 10H]$^{10+}$, **e** [M + 11H]$^{11+}$, **f** [M + 12H]$^{12+}$, in the temperature range of 295–190 K with colour shading to denote each temperature; Error bars represent the standard deviation from three replicates. Here is the interactive plot for data above.

frozen compact conformational population (denoted by an asterisk * in Fig. 3d) at 250, 210, and 190 K. This compact species (*) is not resolved at higher temperatures, indicating that it arises from a smaller conformer that unfolds at higher temperatures. The rate of this conformational transition must be much higher at higher temperatures, with a lower activation barrier, and must occur to completion during the thermalisation in the first few mm of the drift cell, explaining the absence of this structure during those experiments. There is no overlap between this kinetically trapped compact conformation and the restructured conformational ensembles at 250 and 275 K. The [M + 9H]$^{9+}$, [M + 10H]$^{10+}$ and [M + 11H]$^{11+}$ ions (Fig. 5) also exhibit more complex changes at 210 and 190 K. They are highly dynamic and have a broad conformational landscape at 295 K, which resolves to very distinct conformers at 275 K suggesting that the interconversion rate decreases with temperature and states that are freely interconverting at higher temperatures become kinetically trapped. At 250 K these distinct conformers now disappear and - as for β-casein -substantial restructuring is taking place to extended highly overlapped conformers. In summary, the high charge state α-synuclein ions only cold restructure at 250 K and are conformationally trapped at 275 K and below 250 K. To further investigate the behaviour of α-synuclein at 250 K, we performed in source activation, by increasing the cone voltage from 40 to 90 V prior to IM-MS measurements (Supplementary Fig. 9). Whilst we see a slight increase in the overall CCS values for the conformational distributions upon in-source activation, it is far less compared to what we observe for low drift gas temperstures,

especially for the lower charge states than observed at 295 K, indicating that cold restructuring is taking each conformational ensemble to its most extended state. For each of the charge states at 250 K there is conformational occupancy just under 3000 Å², which is also occupied by the activated conformers at 295 K. Remarkably the theoretical CCS limit for the linear sequence is 3880 Å² which suggests that conformers that are either cold activated or in source activated can still retain with some structural elements held rigid by non-covalent interactions, for example helices[58,59]. The effect of cold restructuring on α-synuclein is of a similar magnitude to that achieved in aIMS experiments at ambient temperature (Supplementary Fig. 7, Fig. 9). Take the case of [M + 8H]$^{8+}$ as an example; when the activation energy reaches 600 eV, the $^{TW}CCS^{295K}_{N2→He}$ is ~ 2200 Å², similar to the non-activated $^{DT}CCS^{250K}_{He}$ at 250 K. By contrast, the unfolding of β-casein observed at 250 K is less than the unfolding with 600 eV in the collisional activation experiment. In comparison, as we have previously shown, low charge states of ubiquitin unfold at 600 eV in aIMS but exhibit no significant structural rearrangement during IM-MS measurements at 250 K[26]. These observations illustrate the sequence dependence of cold denaturation even when applied to IDPs.

### Conformational dynamics of α-synuclein captured in real-time: experimental measurement of conformational transition rate constants

The $^{DT}CCSD^{295K}_{He}$ of [M + 13H]$^{13+}$ ions of α-synuclein at 295 K have two highly distinguishable conformations (Figs. 6a, and 5f) with ATDs that

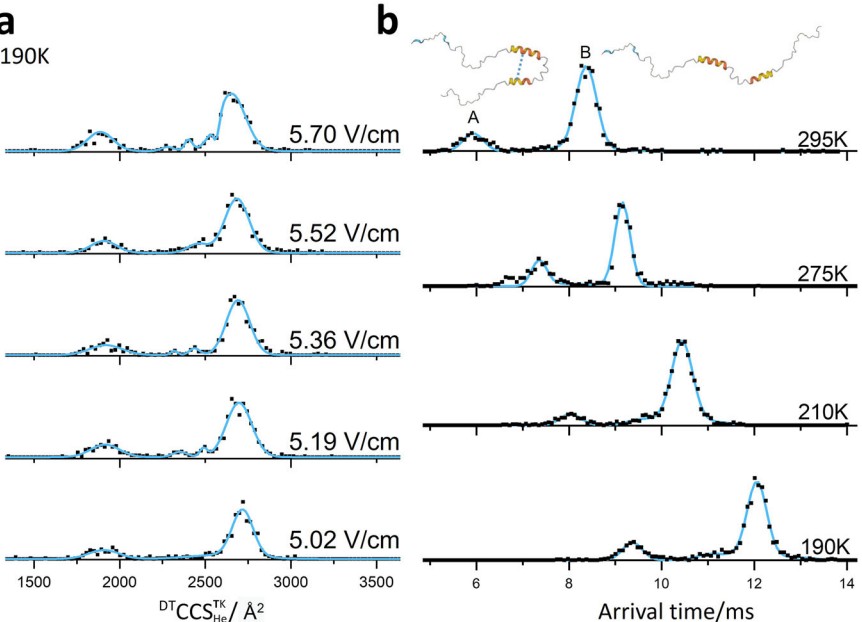

**Fig. 6 | Sub-ambient temperature $^{DT}CCSD190K$ He and ATD of α-synuclein [M + 13H]$^{13+}$. a** Fitted $^{DT}CCSD^{190K}_{He}$ (blue line) and experimental values (black dots) at 5 drift voltages, in decreasing order, at 190 K. **b** Interconversion rate constant calculation simulation fitted to the ATDs obtained at the lowest drift voltage for which reasonable signal was obtained at: 295, 275, 210, 190 K from top to bottom. Black dots represent experimental values, and solid blue lines of panel (**b**) represent the calculated fit to the ATD. The schematic of the α-synuclein structure is adapted from AlphaFold Protein Structure Database[82].

also contain ion density between these two species over the temperature range examined, a behaviour that is characteristic of gas-phase reactions[50]. Since we are examining monomeric molecular cations and we have excluded other sources of such ions, such as dissociating multimeric states, the observed behaviour can be attributed to gas-phase conformational dynamics. From this exemplary dataset we apply the curve-fitting method, modelling the reaction-diffusion-advection phenomena in the drift tube experiment, to obtain kinetic rate constants for those transitions directly by analysing the arrival time distributions and fitting them with a reaction-advection-diffusion model using two reacting species (a similar method has been employed by Poyer et al.[60]). Since the initial and final states are not baseline separated, the conformational transition is not driven to an equilibrium state, where the species would appear as a single peak between the two. When the temperature is lowered to 190 K (Fig. 5a), the rate constant is reduced further, slowing down the reaction. When the drift voltage gradually decreases from 5.70 to 5.02 V cm$^{-1}$, which corresponds to longer drift times, the abundance of compact conformation A decreases. In the first four voltages, we captured some intermediate states between conformations A and B. When the drift voltage reached 5.02 V cm$^{-1}$, the intermediate states are not abundant, and the content of A reached the lowest value among the five groups of voltages. At longer times the process tends more towards an equilibrium state. Similar conditions were not captured for other charge states. Average rate constants from A to B at 295, 275, 210, and 190 K are 0.69, 0.63, 0.50, and 0.47 s$^{-1}$, respectively, which follow an Arrhenius relationship (Supplementary Fig. 12), bolstering the viability of curve fitting. The activation energy, assuming a three-state model, is at 1.79 ± 0.12 kJ/mol from A to B. Intriguingly this activation energy is far lower than expected transitions; and is only equivalent to the formation or breaking of one hydrogen bond. We can speculate on what conformers could have such a small energy difference.

Martin F. Jarrold and co-workers have reported the unfolding of an desolvated helix upon experiencing a similar temperature change[50], with a rate constant for this of 243 s$^{-1}$. There is ample evidence for preservation of helical proteins and helices in vacuo, and we have previously reported that some peptides will form helices upon transfer to the gas phase[61]. There is also evidence that α-synuclein has a helical forming propensity[62,63], which is supported by AlphaFold predictions (Fig. 7) albeit only in a few regions or induced by TFE[64], and so likely only transient in aqueous environments for this highly disordered protein; in-cell NMR has not revealed any evidence for substantial helicity[65]. It has been suggested that α-synuclein may form a more stable helical structure in a highly hydrophobic environment[63,66,67]. Given that the gas phase is the ultimate hydrophobic environment, we speculate that both conformers may contain helical regions formed or captured upon desolvation, and that these have formed a helix-helix interaction (early arriving conformer) which is stabilised by 1 or 2 inter helix H bonds, which break to 'pop' a hair pin to a more extended form (Fig. 6b). A β-hairpin that constrains some transient helical regions has been postulated by MD simulations[68], and metal ions and the molecular tweezer CLR01 have been shown to dramatically alter the conformational landscape of α-synuclein[69]. These studies, along with the finding here are all indicative that distal regions of this protein can be held together by small interfaces in the *apo* form that, when disrupted, induce a large conformational change. The observation that these configurations survive even at 350 K, indicates that both possess some inherent stability as would be imparted by secondary structural elements.

## Discussion

Variable-temperature ion mobility spectrometry has allowed us to map out the conformational landscapes of three different proteins and contrast the behaviour of these with that of a synthetic G5 PLL dendrimer. For the proteins we studied, VT IM-MS showed directly that in addition to a temperature dependent glass transition in polypeptides, cold denaturation is accompanied by marked conformational changes. These effects are not present in the rigid G5 PLL dendrimer and become more pronounced in intrinsically disordered proteins. Figure 7 summarises the major differences between the effects of making VT IM-MS measurements on the rigid dendrimer and on the IDPs. For the dendrimer (Fig. 7a) the behaviour is as predicted by ion transport

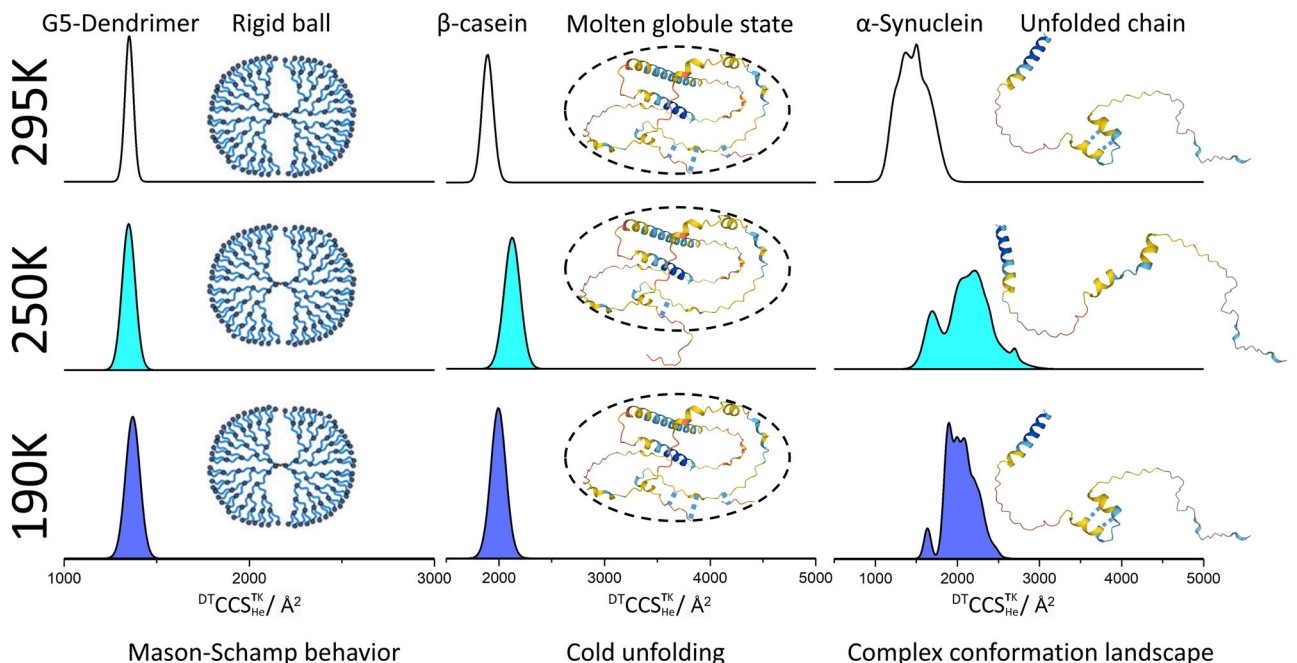

**Fig. 7 | Summary of the effect of drift gas temperature on the conformational landscapes of the G5-dendrimer, β-casein and α-synuclein with representative CCS distributions from the most abundant charge states measured (see above).** Illustrative structures using AlphaFold Protein Structure Database[82] act as a starting point for schematic representations to illustrate candidate geometries for β-casein and α-synuclein and our previously reported schematic for the G5-dendrimer[55]. The blue dotted lines represent possible stabilising hydrogen bonds that are disrupted at temperatures around 250 K. At each of the experimental temperatures of 295, 250, and 190 K, each charge state of the G5-Dendrimer structure is globular and does not restructure, and this model system behaves as predicted by the Mason-Schamp relation[5]. β-casein mostly also only presents one conformer per charge state, which probably are molten globule states, stabilised by many noncovalent interactions, at 250 K hydrogen bonds are lengthened and the protein ion still has residual energy, so these stabilising interactions are disrupted. α-synuclein has a far more complex conformational landscape even within each charge state, which may be part explained by transient helical regions stabilised in the gas phase. Any hydrogen bonds will also lengthen and potentially break at 250 K.

theory, and the increase in CCS as the temperature decreases does not indicate any restructuring of the analyte. This is similar to the effect of taking measurements at lower drift gas temperatures for the native charge states of the highly structured protein ubiquitin; and as such we can surmise that when complexes are rigid and held by many stabilising non-covalent interactions, they will not alter their shape substantially during the ~5-20 ms time spent in the cold drift gas. The higher charge states of ubiquitin as reported here, do appear to alter in conformation at 250 K, and this indicates that the fewer number of stabilising non-covalent contacts are not sufficient to hold the more extended structure together at a temperature when the H-bond lengthen and there is still enough residual energy in the analyte to allow these to break. Previously, we have shown similar restructuring at lower drift gas temperatures for BTPI, cytochrome c, myoglobin at 260 K and lysozyme and IgGs at 250 K[27,29]. For lysozyme, the CCS increase at this temperature is greater in its disulfide reduced form, which also underlines the importance of stabilising interactions and the subtle way the conformational stability can be altered at sub-ambient temperatures.[26]

For IDPs and denatured forms of ubiquitin, varying the temperature at which the IM measurements are carried out provides insights to the low barriers for interconversion between conformers for these conformational dynamic molecules. For β-casein, the protein is mostly only present as 1 or at the most 2 conformers per charge state, and all of these extend at 250 K, particularly for the higher charge states. For α-synuclein the VT IM-MS measurements show a complex interplay between the ability of the measurement to resolve different conformers and how these are differently susceptible to restructuring between 225 and 275 K.

From an energetic point of view, the unfolding behaviour at 250 K is the result of the competition between molecular kinetic energy and

potential energy. At a certain temperature, the decrease in thermal motion is balanced by the increase in potential energy. The molecule requires a certain residual thermal energy to overcome the stabilising effects of the weakening H-bonds to restructure to a more extended conformation (Supplementary Fig. 11). At lower temperatures the extent of residual intramolecular motions is lower and the freezing process is faster which means that the molecules appear to retain their ambient temperature conformations. The $^{DT}CCS^{TK}_{He}$ values obtained at 210 and 190 K are consistently lower than those measured at 250 K. Comparing the two IDPs low-temperature data above, we are led to conclude that the energy barriers in the α-synuclein conformational energy landscape are lower and it is more able to extend. Comparatively, structured proteins have much higher energy barriers due to a greater number of intramolecular interactions, thus limiting the extent of structural change at 250 K. Interestingly the magnitude of conformational change for α-synuclein at 250 K is larger than that experienced in aIMS experiments, but for β-casein the opposite is observed. For the single example of interconversion between a compact and an extended conformer of α-synuclein shown, the energy barrier appears very small, which highlighting the power of sub ambient IM-MS experiments in discerning the low energetic differences in the conformational landscapes of these disordered proteins.

There are several areas in which the contribution of gas-phase measurements on protein structure as a function of low temperatures can inform biology and biophysics[70]. Firstly, there is intrinsic fundamental interest in the behaviour of proteins in the absence of solvent. The full phase space of protein systems, including those of abiotic conditions, is worthy of investigation. Such measurements can be used to inform simulations and to refine force fields which tend to be parameterised only on 0 K data and subsequently coupled to given temperatures. Secondly, cooling or exposure to a vacuum is routinely

used in many biophysical measurements. Electron microscopy (e.g. TEM and SEM) typically uses a combination of cryogenic temperatures and a vacuum[71]; NMR and EPR often freeze samples prior to analysis; protein crystals are also often frozen before diffraction experiments[72]. Understanding the behaviour of isolated proteins allows us to distinguish between solvent effects and effects due to the intrinsic physical properties of these molecules. Thirdly, there are multiple real-life applications that use conditions that approach those used here. Proteins and peptides are constituents of pharmaceuticals, food, and cosmetics, and in all those cases they may be subjected to conditions akin to those explored here, such as freeze-drying for processing or storage. The influence of such conditions on polypeptide conformations can be relevant to every one of these sectors. Finally, in terms of low temperatures alone, diverse psychrophilic organisms exist, including ice algae[73,74], and it will be interesting to investigate systematically in the future how cold denaturation depends on amino acid composition. As for the relevance of gas phase measurements, we must keep in mind that conditions in vivo are very different to the dilute solutions often used in biochemistry. A protein in the cytosol does not contain many water solvation shells before encountering another biomolecule, including amphipathic or hydrophobic molecules. Transmembrane proteins are functional when surrounded by lipids and not polar molecules[75,76]; in this case the low dielectric constant of the vacuum ($\varepsilon_0 = 1$) is much closer to that of lipids ($\varepsilon \approx 4$) than for water ($\varepsilon \approx 80$)[77].

Overall, high-resolution IM-MS can separate and characterise the structures of conformational heterogeneous molecules, such as IDPs, which, due to this heterogeneity, cannot be characterised adequately by methods relying on bulk measurements.

The benefit of having temperature as a variable for IM-MS experiments allows us to demonstrate clear differentiation between the behaviour of conformationally dynamic proteins (IDPs and denatured forms) and more rigid molecules due to their covalent and non-covalent interactions. The distinct restructuring observed at drift gas temperatures close to 250 K appears to be a universal property of naturally occurring proteins taking place at the temperature at which they are predicted to cold-denature[26,27,29]. Such insights, which are hard to obtain with any other experimental method, are valuable for understanding the fundamental effect of temperature on protein stability which may also have practical uses in informing on the storage of proteins and in examining critically the effect of temperature on the proteins in plants and other food supplies as we experience the ongoing climate emergency.

## Methods

### Sample preparation
A poly (L-lysine) (PLL) dendrimer of 5th generation was provided by AstraZeneca. Ubiquitin and β-casein were purchased from Sigma Aldrich, UK. Human recombinant α-synuclein was a gift from Rajiv Bhat, Jawaharlal Nehru University, expressed in BL21 (DE3) Escherichia coli and purified[78]. Ammonium acetate was purchased from Fisher Scientific (Loughborough, U.K.). Final protein concentrations were prepared to 20 μM β-Casein, 20 μM α-synuclein in 50 mM ammonium acetate, pH 6.8. Ubiquitin was prepared in $H_2O$:Methanol:formic acid = 49:49:2, pH 3.0. Stock solutions for each sample are stored in a −80 °C freezer and are thawed 2 hours before each experiment. In our earlier exploratory study of α-synuclein we used a higher concentration and less stringent conditions regarding sample storage prior to analysis[26]. Representative mass spectra for each analyte are presented in the Fig. 2c, along with biophysical data and predicted mass for each protein, the value for the amount of net charge that could be supported on the surface of the isolated protein if it was globular[48]. Supplementary Fig. 1 represents the sequence, the Kappa value ($\kappa$) in Supplementary Table 6 gives an indication of the segregation of charge[45,79].

## Variable temperature ion mobility mass spectrometry
We employed a home-built instrument, which can be operated over the temperature range 120–520 K to obtain all measurements[57]. The configuration of this instrument and the methodology applied to obtain measurements as a function of different drift gas temperatures has been described in detail previously[26,29,57]. In brief: samples are loaded into fused silica nano-ESI capillaries (World Precision Instruments) into which is inserted a platinum wire (Goodfellow). These tips are loaded into the Z-spray source of the mass spectrometer, equipped with a sampling cone interface, and the analyte ions are subsequently guided through two stacked ring RF guides to a trapping region prior to the drift tube. The drift tube consists of a series of electrodes, over which we can apply a field of 4.95-6.14 V cm$^{-1}$. It is encased by a glass tube on the outside of which are heaters and a cooling coil[57]. The drift tube is filled with helium and its pressure is measured with a baratron (MKS instruments) and held constant to ±0.02 Torr for each set of measurements. The drift cell is heated with wire-wound-ceramic heaters and cooled via the ingress of nitrogen gas through a serpentine copper coil that surround the drift region. This nitrogen gas is cooled by passing through a liquid nitrogen tank. For any given measurement the temperature is monitored on the outside with two K type thermocouples and with the drift cell with two PT100 thermometers. For each set of measurements, the temperature of the drift gas is maintained to ±2 K. For each measurement the E/N ratio is maintained as low as possible to prevent alignment of the ions in the drift field and to ensure a similar number of collisions at each different temperature. Practically, the E/N values are all below 10.4 Townsends (Td.) (Supplementary Table 7). which is far lower than that found in commercial IM-MS instruments and corresponds to kinetic energy imparted by the drift field that is lower than the kinetic energy due to thermal motions $k_bT$ ensuring that no field heating occurs and that ions are not aligned in the field during mobility separation[49,80]. The capillary voltage is typically held at 1.1−1.4 kV and the source temperature is set to 80 °C. Inspection of the mass spectra at different temperatures (Supplementary Fig. 2) indicates that cooling of the He buffer gas of temperature does not alter significantly the charge state distributions (CSD) of all analytes examined. For each ion of interest, at each of the charge states it presents (Fig. 2c), we record an arrival time distribution (ATD) which we convert to a CCS distribution, at $T = 190, 210, 250, 275, 295$ and 350 K.

### Reporting summary
Further information on research design is available in the Nature Portfolio Reporting Summary linked to this article.

## Data availability
The mass spectrometry proteomics data have been deposited to the ProteomeXchange Consortium via the PRIDE[81] partner repository with the dataset identifier PXD061503. Unless otherwise stated, all data supporting the results of this study can be found in the article, supplementary, and source data files. Source Data are provided with this paper. The source data used in each figure in the main text and supplementary is provided as a zip file entitled Source Data which contains the data for the figures in the main text 1 and supplementary 2 respectively. Where relevant we have included interactive plots as links in the main text. The code and.csv files for the Arrhenius fits can be found in Supplementary Data 1. All spreadsheets used to convert arrival time distributions to collision cross sections and RAW MS data are also available at FigShare [https://doi.org/10.6084/m9.figshare.25145867] Source data are provided with this paper.

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

## Acknowledgements

We acknowledge the support of EPSRC through the strategic equipment award EP/T019328/1 and for the Prosperity Partnership award EP/S005226/1, and BBSRC for the award of a strategic longer and larger grant GENPENZ BB/X003027/1, the European Research Council for funding the MS SPIDOC H2020-FETOPEN-1-2016-2017-801406 and Waters Corporation for their continued support of mass spectrometry research within the Michael Barber Centre for Collaborative Mass Spectrometry. F.B. acknowledges the Department of Chemistry and AstraZeneca for funding a strategic CASE studentship. The authors also thank the staff in the MS and Separation Science facility, in the faculty of Science and Engineering for their assistance. E.N. is grateful for funding through a University of Manchester Alumni Impact award and from Bristol Myers Squib.

## Author contributions

X.W.: Design of experiment; data collection, processing and analysis; drafting, editing of article. J.D. and J.M.D.K.: Data analysis and interpretation. E.N., J.D., F.B., and J.M.D.K.: data collection and editing of article. R.M.E. and T.B.: Sample preparation. P.E.B.: supervision of project, data analysis and interpretation; design of experiment; editing of article. All authors contributed to the final submitted draft.

## Competing interests

R.M.E. and T.B. are both employees of AstraZeneca and own or have the option to own stocks in this company. The remaining authors declare no competing interests.
