## [Transparent Peer Review file · Nature Communications]

Conformational landscapes of rigid and flexible molecules explored with variable temperature ion mobility-mass spectrometry

Corresponding Author: Professor Perdita Barran

Version 1:

Reviewer comments:

Reviewer #1

(Remarks to the Author)

The manuscript by Wang et al describe the use of variable temperature ion mobility mass spectrometry (VT-IM-MS) to characterise structural stability of gas-phase ions. Part of this work is aimed at understanding the thermal stability of gas-phase conformers.

Four monomeric, model protein systems are selected for this study. However, (most of) these systems have been extensively studied by this group and others, including using VT-IM studies. This raises the concern of novelty, which (at present) reads as a continuation of prior work but on different model systems. What is new? How do these results inform on solution chemistry (see point below)?

A larger concern is the fact that these VT studies are performed in the gas-phase – life requires water. This greatly diminishes the significance of the presented results, restructuring of “desolated” ions in a drift tube at different temperatures. In addition, results for temperature-dependent changes in CCS are over interpreted without any supporting data. For example, “leaves conformers with some structural elements held rigid by non-covalent interactions”. CCS does not provide such insight into the atomic structure(s) of gas-phase ions. These remarks should be moved from results to discussion.

In the abstract describes “free energy landscape of these proteins”. No free energy (or thermodynamics) is provided. The paper reads as a mere observation of results. The paper is presented in technical form (largely description of observables). Whilst the VT-IM results are interesting and rigorously performed, these results are most appropriate for a journal in the mass spectrometry field.

Reviewer #2

(Remarks to the Author)

In this manuscript, the authors build on their previous work with variable-temperature ion mobility spectrometry to distinguish the effects of temperature on ion-neutral interactions and on gas-phase ion conformations. They report different behaviour for rigid structures, structured proteins, and disordered proteins, and find evidence for temperature-dependent conformational changes in alpha-synuclein that seem to be due to the disruption of a single hydrogen bond. Overall, I believe this is an exciting and potentially very significant biophysics paper, although I do have a few comments the authors may wish to consider.

I find the explanation of why the collision cross-section increases with decreasing temperature in the first section of the Results somewhat confusing. The authors state on lines 208-212 that for a rigid structure like a dendrimer, a decreasing CCS with increasing temperature is predicted by the Mason-Schamp equation, and that CCS should be inversely proportional to the square root of the temperature. However, looking at Equation [1], this is only true if the electric field, buffer gas density, and drift velocity are all kept constant. Obviously, keeping E and N (but cf. infra) constant can be done, but I would not expect the drift velocity of ions to remain constant. Also, on line 211, the authors write that a temperature decrease from 290 to 195 K induced a CCS increase of 1.3%, whereas an inverse square root relationship should result in a 22%

increase. I suspect, therefore, that the drift velocity and/or buffer gas density at lower temperature were significantly higher, but that after solving Equation [1], a 1.3% increase in CCS was found, which – as the authors discuss on lines 46-50 – is likely due to long-range interactions between the buffer gas and analyte ion becoming more pronounced at low temperatures. This behaviour, however, is not predicted by the Mason-Schamp equation, since that assumes a hard-sphere model. I would suggest that the authors rewrite this section for clarity. In fact, it might be good to show some arrival time distributions in the Supporting Information to accompany the CCS distributions shown in the main text. To be clear, I do not believe that any of this affects the key results or their interpretation, but the explanation should be clarified.

Related to this, in the Methods, the authors state that E/N values were kept below 10.4 Td, but it is not clear to me whether they adjusted the pressure to keep the number density N constant between experiments at different temperatures, or kept the pressure constant and had significantly different values of N between experiments. The statement that 'pressure is [...] held constant to ± 0.02 Torr for each set of measurements' implies the latter, although 'set of measurements' is not completely unambiguous. It is also not fully clear to me how the authors measured N. If this was calculated based on measurements of pressure and temperature, did they take non-ideal gas behaviour into account? I would suggest that the authors clarify these points, and that they add a table in the SI with the electric field strength, pressure, and buffer gas number density for each measurement.

On a related note, the statement 'the E/N values are all below 10.4 Townsends (Td.) which is far lower than that found in commercial IM-MS instruments and is significantly lower than kbT ' is odd, as Td has the dimensions of electric field strength over volume, while kbT is energy. I suspect that the authors mean that the energy imparted by the electric field is negligible compared to the thermal energy of the ions, but this should be phrased more clearly.

On lines 230-232, the language is somewhat ambiguous. The sentence 'This is beautifully exhibited by the behaviour of the G5 dendrimer, which exhibits a small decrease in CCS when the DT temperature is increased to 350 K, in line with kinetic theory (Equation [1]).' could be interpreted as meaning that at high temperatures, the behaviour becomes more hard-sphere-like and is then modelled by Equation [1] (which is presumably what the authors intended) but it could also be interpreted as meaning that the transition from a regime in which long-range interactions are important to hard-sphere-like behaviour is predicted by Equation [1], which would be inaccurate since that equation assumes hard-sphere behaviour, as discussed in a previous point.

On line 304 and again on line 373, the authors refer to the combination of collisional activation and IM measurements as 'aIMS'. However, in the SI (S5 and S6) as well as on lines 369 and 472 of the main text, the more common 'collision-induced unfolding' or 'CIU' is used. It would be good if the authors could either use consistent terminology if the two terms are meant to be synonymous, or briefly explain the difference if they are not.

Minor points:

On line 37, there is a word missing in 'under the influence of weak electric field'.

On lines 44-45, the authors discuss the work by Bowers and colleagues on temperature-dependent IM of C60 clusters and cite [Science 267, 1483–1485 (1995)]. Can the authors check that this is indeed the reference they intended to cite for this? It seems to me that there are other papers published by the Bowers lab around that time that are more relevant to this part of the text.

Line 112-113: Grammar issue in 'we intriguingly found that an increase in the CCS at 260K indicating that the proteins'.

Line 135: 'conformational dynamic'  'conformationally dynamic'

Line 172: word missing in 'We employed a home built instrument, which can be operated over 120-520 K obtain all measurements.'

Line 175 and SI page 12: Inconsistent spelling of 'z-spray' vs. 'Z-spray'.

Lines 266-267: the authors write that 'The majority of living organisms on earth, need to survive temperatures between 250 and 350K.' This seems like a rather extreme range to apply to 'the majority of living organisms on Earth'.

Lines 366-367: The phrase 'which suggests that cold activated or in source activated still leaves conformers' is odd.

Line 369: 'of a similar magnitude to than achieved in CIU experiments' – I suspect 'than' should be 'that'.

Line 372: 'low charge states of ubiquitin unfolds' – 'unfold'

Line 411: 'alpha-fold'  'AlphaFold'

SI page 9: a sentence fragment 'the way these' appears seemingly for no reason at the start of the text on this page.

SI page 10: '[References]' was presumably meant to be deleted (or replaced with actual references) before submission.

The links to the interactive plots do not seem to work.

Reviewer #3

(Remarks to the Author)

The manuscript by Wang et al. Details the use of a variable temperature ion mobility drift cell to probe the conformational landscape of select protein charge states in the gas phase.

There are places where the language could be refined and references cited in more detail. Nevertheless, the data from these experiments is compelling and of broader interest, despite the use of a highly customized instrument. The majority of analytical measurements are conducted on an ensemble of analytes with the dilution being how finely these subpopulations can be evaluated. A key point that could strengthen the manuscript is the notion that IM-MS can provide detailed information on subpopulations that, compared to solution phase approaches, are quite small. This work provides key insights into the conformational landscape that is available to the proteins evaluated. Given the lack of extensive low-temperature data on gas-phase protein species and how temperature may impact observed conformations, there is merit in the publication of this manuscript.

This enthusiasm is tempered to a small degree by the desire to see additional quantitative details regarding the transitions observed. Additionally, the authors are encouraged to provide details, when possible, for alternative explanations that may contribute to the range of conformations observed for gas-phase ions.

The authors make it a point to quantify changes in the peak width with respect to diffusion, temperature dependent mobility effects, and contributions from conformation changes. These are key points and the authors are encouraged to provide additional detail on how these distinctions are made.

Other overarching concerns that the authors are encouraged to recognize, and dismiss if warranted, include whether the observed species across charge states could be derived either through charge reduction during the drift cell or larger oligomers. Further, because the temperature is lowered are there any scenarios where solvent adducts exist within the drift cell but are removed prior to mass analysis that might stabilize select conformations?

The use of the dendrimer as a control is a clever approach that helps bound the scope of interpretation. One aspect of the G5 dendrimer is the observation that the peak width actually increases as temperature decreases. This is not what one might expect. Can the authors please address this behavior?

For aSyn data, there is a considerable degree of conformational variation explored by the protein, even across charge states. In fact, there are some distributions (e.g., +7 @250K) that are quite narrow. How do these peaks compare to the theoretical minimum predicted by diffusion and how do these compare to G5? The point of this question is frame the range of different conformations that exist within the larger distributions.

Across the data presented (including the bar chart) the data for 275K shows a considerable degree of conformational flexibility for the protein charge states probed. It is unclear exactly the order of operations for the experimental analysis campaign but is there something special about 275K, was there an experimental differentiator at that temperature, or is this observation simply coincidence?

While asking for additional data is somewhat contentious, the work by Poyer et al. probing gas phase isomerism provides an interesting framework to quantify changes from a thermodynamic perspective. Is there any chance some of the species probed are in some sort of thermal equilibrium that can be captured on the experimental timescale of the ion flight time or are the species observed reasonably static? Is there a way to thermodynamically quantify transitions between temperatures?

Below are specific comments related to the text:

Abstract:

L18 — How do you “provide the role of temperature?”

L19 — There are a variety of collision effects. How do you differentiate between the range of types (e.g., collisional cooling/heating, dissociative, etc.)?

L23 — Should read: “does not alter significantly over this temperature range.”

Introduction

The authors should make it clear that the Mason Schamp equation is a first approximation at best. Including the concept of a momentum-transfer cross-section would also help bound expectations for the veracity of this equation.

L42 — ExellIMS does sell variable temperature drift cell but it is aimed squarely at the high temperature range.

L82 — Should read: “requirements of such an instrument,”

L84 — A reference should be included on this line.

L136 — Why are the proteins examined “exemplar?”

Instrumentation

Are the authors confident that the ions reach thermal equilibrium prior to injection into the mobility cell? The “ion buncher” is quite short in the manuscript referenced by the authors. Essentially, making a statement related to concept that the ions are thermalized to the experimental temperature prior to injection would be helpful.

L230 — Can the authors use a different word than “beautifully?”

Version 2:

Reviewer comments:

Reviewer #1

(Remarks to the Author)

The authors have addressed the concerns of my initial critique. I recommend for publication.

Reviewer #2

(Remarks to the Author)

The authors have significantly improved the manuscript with this revised version. I only have a few minor points that they might want to consider.

The revised text in the Introduction does a better job of explaining the two different effects of temperature on measured CCS values assuming no structural changes. The claim (made in the Introduction and repeated on line 227) that CCS is expected to scale with the inverse square root of temperature might still not be obvious to the non-specialist reader though, as it relies on the assumption that the product of these two values remains constant. As the authors already stated in their reply, the validity of this assumption depends on how the momentum-transfer integral changes as a function of temperature, and on whether the long-range polarisation potential is the dominant factor for interactions. This seems realistic under the low-temperature conditions used in this work, and the text alludes to this assumption by stating that long-range attractive forces dominate at low temperatures. However, I believe it would be useful to add one or two sentences to emphasise this point in a bit more detail in order to avoid potential confusion (perhaps even introducing the concept of a momentum-transfer cross section as was suggested by Reviewer 3) as the text near the end of page 3, and the positioning of Equation 1, currently might give the impression that this correlation can be derived from the Mason-Schamp equation. That said, this is a fairly minor point that does not detract from the significance of the work, and I fully agree with the authors that temperature-dependent structural changes are the most interesting effect studied in this work, and that there is a clear difference between the behaviour of systems where such structural changes are limited (such as the dendrimer) and those where they are more pronounced.

In Figure 3, very broad CCS ranges with a lot of whitespace are shown, in particular for the dendrimer, ubiquitin, and beta-casein. This makes it difficult to see details of the distribution as well as small CCS differences between measurement temperatures. Providing links to interactive plots for the reader to explore is nice, but perhaps the distributions in the figure could be zoomed in a bit.

Minor points:

On line 49, the fragment ‘giving rise to the observed a greater CCS values’ has grammar issues. On the line before that, ‘the collision trajectory of the ion-neutral pair is farm more influenced’ has a typo.

On line 280, the authors mention ‘sharp conformers.’ The signals associated with the conformers are sharp, not the conformers themselves.

Reviewer #3

(Remarks to the Author)

Based upon the response to the reviewers, there appears to be some concern related to the novelty approach and the relevance of the gas phase approaches. While the point of Reviewer #1 with respect to water is noted, the points made by the authors are salient. While it is true that the impact of bulk water is largely removed, an extension of Reviewer #1's concern to other techniques would also preclude the rigorous interpretation of biological thermodynamics using those approaches--a crystal structure, by definition, is derived from non-native conditions. Overall, the authors have provided detailed responses to most concerns, emphasizing the novelty of their findings, refining the clarity of their explanations, and making necessary revisions to improve the manuscript.

There are a few points worth noting that could further aid in developing the manuscript. Aside from the estimation of CCS, perhaps the most quantitative work presented is the data from the Arrhenius plot in the Supp Material. The authors state that

the parameters for this plot are derived from a fit of Equation 2 in the Supp Material to the experimental data. Moreover, they noted that, "The blue solid line presents a good fit to the experimental ATD." However, these fits are not shown. Are they suggesting that the light blue line shown in Figure 6 of the main manuscript are those fits? If so, this is not clear in the manuscript. Finally, given the need to clearly define the methods and approach, perhaps the authors should simply show these fits that go into the Arrhenius plot and release the CSVs and associated code contributing to those fits. This would further support the method they aim to expand upon and alleviate some of the concerns raised by Reviewer #1.

With minor clarifications, this manuscript warrants publication as there are few experimental approaches that can provide detailed insights that complement the more common interpretations derived from the canonical averages derived from NMR, crystal structures, and microscopy techniques.

Minor Comments:

Spacing between reported values and units should be included (e.g., 273 K and not 273K).
m/z is italicized.

Typo in the supplemental material: "which equal to $1-t_A/t_B$, β and θ "

Version 3:

Reviewer comments:

Reviewer #2

(Remarks to the Author)

The authors have done a good job addressing my comments. I believe the manuscript in its current form is worthy of publication.

Reviewer #3

(Remarks to the Author)

The authors appear to have addressed all of the major points made by the reviewers. With the inclusion of the data in the supplemental material and the clarification of the points related to the CCS concepts, the manuscript appears ready for publication.

Reviewer comments

We are grateful to all the reviewers for reading carefully our manuscript and providing valuable suggestions and scrutiny on the work presented in the current paper. Your comments have guided us in making significant revisions to the manuscript in order to address them. We are glad that we can respond to the major points each reviewer has raised.

Reviewer #1 (Remarks to the Author):

The manuscript by Wang et al describe the use of variable temperature ion mobility mass spectrometry (VT-IM-MS) to characterise structural stability of gas-phase ions. Part of this work is aimed at understanding the thermal stability of gas-phase conformers.

1. *Four monomeric, model protein systems are selected for this study.*

However, (most of) these systems have been extensively studied by this group and others, including using VT-IM studies. This raises the concern of novelty, which (at present) reads as a continuation of prior work but on different model systems.

What is new?

As described in the manuscript introduction in this work we investigate the effect of lower drift gas temperatures on the observed conformations of ions. Our previous studies on a range of proteins has demonstrated the occurrence of an unfolding event in the 250-280K range^{1,2}. We chose to investigate this effect more systematically, as detailed on page 6, and to consider how the intrinsic, solvated structure of any given protein may be influenced by lower temperatures. For this reason, model systems are chosen, where there is a substantial body of work from solution and gas phase methods on the conformational landscapes. We note that for the proteins chosen there is no existing low temperature IM-MS data (for Ubiquitin we have previously published data on the lower charge states but not on the higher charge states including the $[M+10H]^{10+}$ which has been assigned to the 'A' state). we remind the reviewer that there is no other experimental method that could study the effects of sub 270K temperatures on the structure of proteins.

In summary the novelty of this work is

- (a) the observation of the effect of low temperature on the conformational landscape on intrinsically disordered proteins;
- (b) the observed dependence of this effect on the conformational heterogeneity of those IDPs, as demonstrated by comparing data from Ubiquitin, with that found for alpha-synuclein and beta-casein;
- (c) the clear demonstration that VT-IMMS is a technique that can populate the phase diagram of proteins in a region of low temperature and pressure which cannot be accessed by any other technique with such resolution.

The paper is originally intended for a special issue with the theme of *Collection: Methods and Applications in Multiscale Structural Biology* and for that reason we use systems well-studied in

biophysics and structural biology to demonstrate the reach of the methods. Nevertheless, the observation of cold restructuring as a general phenomenon for IDPs as well as structured proteins is novel, and the use of a very rigid PLL5 dendrimer as a control system to compare with proteins to justify this is also novel.

2. How do these results inform on solution chemistry (see point below)?

A larger concern is the fact that these VT studies are performed in the gas-phase – life requires water. This greatly diminishes the significance of the presented results, restructuring of “desolated” ions in a drift tube at different temperatures.

There are several arguments as to how gas-phase measurements inform biology³. We outline these below and have added these explanations to our manuscript (conclusions and outlook) to assist the reader as to the potential of our method and the findings described herein.

First, there is intrinsic fundamental interest in the behaviour of proteins in the absence of solvent. The full phase space of protein systems, including those of abiotic conditions, is worthy of investigation. Such measurements can be used to inform simulations and to refine force fields which tend to be parameterised only on 0 K data and subsequently coupled to given temperatures.

Secondly, cooling or exposure to a vacuum is routinely used in many biophysical measurements. Electron microscopy (e.g TEM and SEM) typically uses a combination of cryogenic temperatures and a vacuum⁴; NMR and EPR often freezes samples prior to analysis; protein crystals are often frozen before diffraction experiments⁵. Understanding the behaviour of isolated proteins allows us to distinguish between *solvent effects* and effects due to the *intrinsic physical properties* of these molecules.

Thirdly, there are multiple real-life applications that use conditions that approach those used here. Proteins and peptides are constituents of pharmaceuticals, food and cosmetics, and in all those cases they may be subjected to conditions akin to those explored here, such as freeze-drying for processing or storage. The influence of such conditions on polypeptide conformations can be relevant to every one of these sectors.

Finally, in terms of low temperatures alone, diverse psychrophilic organisms exist, including ice algae^{6,7}, and it will be interesting to investigate systematically in the future how cold denaturation depends on amino acid composition. As for the relevance of gas phase measurements, we must keep in mind that conditions *in vivo* are very different to the dilute solutions often used in biochemistry. A protein in the cytosol does not contain many water solvation shells before encountering another biomolecule, including amphipathic or hydrophobic molecules^{8,9}. Transmembrane proteins are functional when surrounded by lipids and not polar molecules; in this case the low dielectric constant of the vacuum ($\epsilon_0=1$) is much closer to that of lipids ($\epsilon\approx 4$) than for water ($\epsilon\approx 80$)¹⁰.

As for the relevance of mass spectrometry methods in general, it is an established result that gas phase ions retain a “memory” of their solution conformation inside the mass spectrometer see for example Robinson *et al.*¹¹, Barran *et al.*^{12,13} Coon *et al.*¹⁴ and even of retained activity Cooks *et al.*¹⁵ This can be understood from first principles from the fact that at room temperature, the energy of formation of a single hydrogen bond (≥ 6 kJ/mol) is greater than kT (2.8 kJ/mol at 300K) by a factor of at least 2. In the presence of water, hydrogen bonds with water molecules compete with intramolecular hydrogen bonds, however in vacuum only 1-2 hydrogen bonds may spontaneously break at all. Further rearrangements require some energetic compensation, *i.e.* the immediate

formation of different intermolecular hydrogen bonds. The availability of those interacting groups creates a major geometric constraint in gas phase structures. This is another reason why the results of our study are surprising, since we have observed structural changes in conditions when few such changes are generally expected. The degree of conformational heterogeneity due to this effect tracks the intrinsic conformational heterogeneity of each protein sequence.

The effects of solvent conditions on protein structure in the gas phase has been the topic of multiple studies, and observations such as solution temperature¹⁶ solvent and denaturants¹⁷, pH¹⁸, salt and cosmotropic additives^{19,20} have been observed by MS and IM-MS.

These comments have been included in our “Conclusions and Outlook” section (Page 25) and the introduction.

In addition, results for temperature-dependent changes in CCS are over interpreted without any supporting data.

In this paper we focus on showcasing the unique insights that come from using the low temperature IM-MS technique as applied to selected model protein systems which harmonises with the topic of the issue that we have submitted. No other method can offer insights into isolated biomolecules with a resolution of charge state as we can with variable temperature IM-MS and so we are not clear on what ‘supporting data’ this reviewer may want us to invoke.

For example, “leaves conformers with some structural elements held rigid by non-covalent interactions”. CCS does not provide such insight into the atomic structure(s) of gas-phase ions. These remarks should be moved from results to discussion.

We agree we cannot always pinpoint the residues that are no longer in contact (although we have previously shown with post activation UVPD Black *et al.*²¹ and Bruker and co-workers with ECD see for example ²² which regions are disrupted) however the increase in CCS in the absence of any decrease in mass means that non-covalent interactions must have been disrupted. We precede the statement above with the word ‘suggests’ and have clarified this in the results and discussion.

In the abstract describes “free energy landscape of these proteins”. No free energy (or thermodynamics) is provided. The paper reads as a mere observation of results. The paper is presented in technical form (largely description of observables). Whilst the VT-IM results are interesting and rigorously performed, these results are most appropriate for a journal in the mass spectrometry field.

We thank the author for appreciating the experimental rigour of the study, we are exploring the free energy landscape of these proteins although we do not in the main quantify it. We had written

...conformational variation that gives **qualitative** insights to the effect of temperature on the free energy landscape of these proteins.

assuming that calling the effects **qualitative** would be sufficient to convey our intent, but we have for the sake of clarity replaced ‘free energy’ with ‘conformational’ in our revised text.

In our example for the conformational change observed for the +13 charge state of α -synuclein we do determine the energy difference between two states based on the result of the curve-fitting approach, and so we counter that we are indeed showing that in the gas phase it is indeed possible to determine activation energies for transitions.

Reviewer #2 (Remarks to the Author):

In this manuscript, the authors build on their previous work with variable-temperature ion mobility spectrometry to distinguish the effects of temperature on ion-neutral interactions and on gas-phase ion conformations. They report different behaviour for rigid structures, structured proteins, and disordered proteins, and find evidence for temperature-dependent conformational changes in alpha-synuclein that seem to be due to the disruption of a single hydrogen bond. Overall, I believe this is an exciting and potentially very significant biophysics paper, although I do have a few comments the authors may wish to consider.

1. *I find the explanation of why the collision cross-section increases with decreasing temperature in the first section of the Results somewhat confusing.*

We have added clarification to the introduction (Pages 3-4) and also in the results section to clarify this (Page 11).

The authors state on lines 208-212 that for a rigid structure like a dendrimer, a decreasing CCS with increasing temperature is predicted by the Mason-Schamp equation, and that CCS should be inversely proportional to the square root of the temperature. However, looking at Equation [1], this is only true if the electric field, buffer gas density, and drift velocity are all kept constant.

We have clarified this statement as the known change in CCS with temperature is also affected by the increased dominance of the long-range attractive potential at the buffer gas temperature decreases.

Obviously, keeping E and N (but cf. infra) constant can be done, but I would not expect the drift velocity of ions to remain constant.

We have changed equation [1] to

$$\overline{v_d} = \frac{3 qE}{16 N} \sqrt{\frac{2\pi}{\mu k_B T} \frac{1}{\Omega}}$$

in order to avoid this confusion. Under conditions of stable mass, charge, electric field, drift gas density, the effect of temperature is seen in this equation in the temperature term and also in Ω .

Also, on line 211, the authors write that a temperature decrease from 290 to 195 K induced a CCS increase of 1.3%, whereas an inverse square root relationship should result in a 22% increase. I suspect, therefore, that the drift velocity and/or buffer gas density at lower temperature were significantly higher, but that after solving Equation [1], a 1.3% increase in CCS was found, which – as the authors discuss on lines 46-50 – is likely due to long-range interactions between the buffer gas and analyte ion becoming more pronounced at low temperatures. This behaviour, however, is not predicted by the Mason-Schamp equation, since that assumes a hard-sphere model. I would suggest that the authors rewrite this section for clarity. In fact, it might be good to show some arrival time distributions in the Supporting Information to accompany the CCS distributions shown in the main text.

To be clear, I do not believe that any of this affects the key results or their interpretation, but the explanation should be clarified.

This is a confusion which may have arisen from our prior phrasing. The effect of the gas temperature on the nature of the collision is encompassed within equation 1, the effect of temperature on the CCS is also due to the interaction potential, which is somewhat hidden in equation 1, but can be found in the references now referred to above (page 3). In summary in order to derive CCS from measurement (which is the drift velocity at a given E) it is important to include T the temperature at which the collisions occur. The effect of that temperature on the ion which is the subject of this report is found by comparing the CCS obtained at each temperature.

On a conceptual level, the assumption of Equation [1] is that collisions between one gas molecule and one ion comprise the vast majority of all collision events, so that many-body effects can be effectively ignored, and that the collisions are elastic. Although a hard-sphere potential without other internal degrees of freedom would result in elastic collisions, the same would be true with potentials that become flat at long distances – i.e. at sufficiently large ion-neutral distances there will be conservation of kinetic energy. Thus, the equation is agnostic on the nature of the long range interaction potential. It must be noted that for proteins and other macromolecules, energetic collisions would be inelastic, since they could change the structure, however that can be the topic of another study.

As suggested by this reviewer we have amended the text to clarify this on pages 3 and 4, and we have also submitted all of the raw data including the ATDs along with this paper in our supporting dataset. We address the E/N point below.

- 2. Related to this, in the Methods, the authors state that E/N values were kept below 10.4 Td, but it is not clear to me whether they adjusted the pressure to keep the number density N constant between experiments at different temperatures, or kept the pressure constant and had significantly different values of N between experiments.*

We have added data table S7 (supporting information) to display the range of E/N values for the experiments. E/N was not held precisely constant at all temperatures between measurements due to difficulties in stabilising temperatures and pressures, but there is overlap between experimental datasets. Since the experiment involves taking measurements of arrival time distributions at 5-6 different field strengths E it is impossible for E/N to be constant among all of them; instead the pressure is kept as stable as possible. In addition, the 2016 report on the VT-IM-MS used in this experiment demonstrate the measurements for 1222 m/z of Agilent Tune Mix were performed in two E/N regimes of 3.8 to 5.8 Td and 7 to 16 Td, yielding results of 229.4 Å² and 229.5 Å², respectively. This demonstrates that the E/N ratio within this range (3.8-16Td) has almost no impact on the final calculated CCS result. In our experiment, the E/N ratio was controlled within the range of 5.09 to 10.67 Td.

The statement that 'pressure is [...] held constant to ± 0.02 Torr for each set of measurements' implies the latter, although 'set of measurements' is not completely unambiguous.

At each temperature we take measurements over at least 5 different drift voltages. That consists a set of measurements. It is standard procedure in linear field drift tube experiments.

It is also not fully clear to me how the authors measured N. If this was calculated based on measurements of pressure and temperature, did they take non-ideal gas behaviour into account?

Non-ideal behaviour was not taken into account. We use Helium as a drift gas to minimise non-ideal gas effects. N is calculated from the ideal gas law using experimental measurements of the pressure and temperature: $p = Nk_B T$

I would suggest that the authors clarify these points, and that they add a table in the SI with the electric field strength, pressure, and buffer gas number density for each measurement.

We have added the E/N values (Table S7) and all the other data is contained for each and every measurement in our supporting dataset.

On a related note, the statement 'the E/N values are all below 10.4 Townsends (Td.) which is far lower than that found in commercial IM-MS instruments and is significantly lower than kbT' is odd, as Td has the dimensions of electric field strength over volume, while kbT is energy. I suspect that the authors mean that the energy imparted by the electric field is negligible compared to the thermal energy of the ions, but this should be phrased more clearly.

The reviewer is correct, and we have now rephrased this (page 10 line 204). We mean that the kinetic energy of the gas-ion collisions is does not greatly exceed the kinetic energy of helium atoms due to thermal motions at each temperature.

3. *On lines 230-232, the language is somewhat ambiguous. The sentence 'This is beautifully exhibited by the behaviour of the G5 dendrimer, which exhibits a small decrease in CCS when the DT temperature is increased to 350 K, in line with kinetic theory (Equation [1]).' could be interpreted as meaning that at high temperatures, the behaviour becomes more hard-sphere-like and is then modelled by Equation [1] (which is presumably what the authors intended) but it could also be interpreted as meaning that the transition from a regime in which long-range interactions are important to hard-sphere-like behaviour is predicted by Equation [1], which would be inaccurate since that equation assumes hard-sphere behaviour, as discussed in a previous point.*

We have changed the language to improve clarity.

There are three effects to consider: First is the effect of scattering due to collisions at different thermal energies; these make no statement about the interaction potential, but assume that they are elastic collisions at thermal energies between one ion and one neutral atom and these result to the temperature term in Equation 1. On top of that there is the effect of temperature on the

momentum transfer integral Ω . These are the effects expected in rigid molecules since they happen even when the average geometry of the molecule does not change at different temperature. Finally there are temperature effects on Ω which are caused by temperature-dependent structural changes. It is the last that we consider most interesting to examine. Since the effects of long-range interactions and structure cannot be separated in an experiment, we use a system that is known not to change over that temperature range, the G5 dendrimer, as an experimental control case. We have removed mention of Equation 1 in the manuscript at these lines to avoid this confusion and as stated above have clarified the description of these contributing effects in the introduction.

4. *On line 304 and again on line 373, the authors refer to the combination of collisional activation and IM measurements as 'aIMS'. However, in the SI (S5 and S6) as well as on lines 369 and 472 of the main text, the more common 'collision-induced unfolding' or 'CIU' is used.*

It would be good if the authors could either use consistent terminology if the two terms are meant to be synonymous, or briefly explain the difference if they are not.

The two terms can be used interchangeably, however aIMS is more general since it describes the fact that collisional activation takes place prior to mobility separation. In gas-phase protein ions the restructuring caused by collisional activation is not always “unfolding” since gas phase structures with a greater degree of intrinsic solvation could become accessible, leading to compact structures before unfolding. CIU appears to bias readers towards “unfolding”, implying more extended geometries whereas aIMS avoids this problem.

We have replaced all instances of CIU with aIMS in accordance to the reviewer’s suggestions.

Minor points:

1. *On line 37, there is a word missing in 'under the influence of weak electric field'.*
We have changed to 'under the influence of a weak electric field'
2. *On lines 44-45, the authors discuss the work by Bowers and colleagues on temperature-dependent IM of C60 clusters and cite [Science 267, 1483–1485 (1995)]. Can the authors check that this is indeed the reference they intended to cite for this? It seems to me that there are other papers published by the Bowers lab around that time that are more relevant to this part of the text.*
We agree this was not the best reference, and have included the correct reference [Wyttenbach, T., Von Helden, G., Batka, J. J., Carlat, D. & Bowers, M. T. Effect of the long-range potential on ion mobility measurements. *Journal of the American Society for Mass Spectrometry* **8**, 275–282 (1997).]
3. *Line 112-113: Grammar issue in 'we intriguingly found that an increase in the CCS at 260K indicating that the proteins'.*
We changed this sentence as 'intriguingly we found that an increase in the CCS at 260K indicates that the proteins'.
4. *Line 135: 'conformational dynamic'  'conformationally dynamic'*
Done

5. *Line 172: word missing in 'We employed a home built instrument, which can be operated over 120-520 K obtain all measurements.'*
We changed this sentence as 'We employed a home built instrument, which can be operated over the temperature range 120-520 K to obtain all measurements.'
6. *Line 175 and SI page 12: Inconsistent spelling of 'z-spray' vs. 'Z-spray'.*
Done
7. *Lines 266-267: the authors write that 'The majority of living organisms on earth, need to survive temperatures between 250 and 350K.' This seems like a rather extreme range to apply to 'the majority of living organisms on Earth'.*
We changed this sentence as 'Many living organisms on earth need to survive temperatures between 250 and 350K'
8. *Lines 366-367: The phrase 'which suggests that cold activated or in source activated still leaves conformers' is odd.*
We replaced this phrase with 'which suggests that conformers that are either cold activated or in source activated can still retain.'
9. *Line 369: 'of a similar magnitude to than achieved in CIU experiments' – I suspect 'than' should be 'that'.*
Done
10. *Line 372: 'low charge states of ubiquitin unfolds' – 'unfold'*
Done
11. *Line 411: 'alpha-fold'  'AlphaFold'*
Done
12. *SI page 9: a sentence fragment 'the way these' appears seemingly for no reason at the start of the text on this page.*
We missed the word 'Considering', now we have added it.
13. *SI page 10: '[References]' was presumably meant to be deleted (or replaced with actual references) before submission.*
Done
14. *The links to the interactive plots do not seem to work.*
We have amended this

Reviewer #3 (Remarks to the Author):

The manuscript by Wang et al. Details the use of a variable temperature ion mobility drift cell to probe the conformational landscape of select protein charge states in the gas phase.

1. *There are places where the language could be refined and references cited in more detail. Nevertheless, the data from these experiments is compelling and of broader interest, despite the use of a highly customized instrument. The majority of analytical measurements are conducted on an ensemble of analytes with the dilution being how finely these subpopulations can be evaluated.*

A key point that could strengthen the manuscript is the notion that IM-MS can provide detailed information on subpopulations that, compared to solution phase approaches, are quite small. This work provides key insights into the conformational landscape that is available to the proteins evaluated. Given the lack of extensive low-temperature data on gas-phase protein species and how temperature may impact observed conformations, there is merit in the publication of this manuscript.

We thank the reviewer for the kind remarks. We have extended our Outlook section to underline the impact of IMMS measurements on understanding the phase space of proteins in conditions (such as low pressure and low temperature) that are not accessible by other techniques.

2. *This enthusiasm is tempered to a small degree by the desire to see additional quantitative details regarding the transitions observed. Additionally, the authors are encouraged to provide details, when possible, for alternative explanations that may contribute to the range of conformations observed for gas-phase ions.*

With ion mobility, ion conformations that do not interconvert over the timescale of the experiment, in our case in the order of 10ms, can be separated provided that they have a sufficiently different CCS. Unfortunately, sufficiently similar structures, as well as dynamic structures, cannot be resolved, with the exceptions of the cases we outline. A different method would be required to completely solve structures. In the end, the experiment does use a single dimension, drift time which can be related to CCS, to order a $3(n-2)$ -dimensional conformational space, n here being the number of atoms in the protein ion. Experimental detailed assignments are not yet possible. Molecular dynamics simulations can, in principle, help with the assignment of structures, but there are no efficient methods that we could currently access to sample correctly the conformational landscape of proteins in conditions appropriate to represent our experiment. Moreover, all modern force-fields are parameterised with solution charges in mind, which means that even extensive and computationally expensive simulations would lead to results that would not be secure. To improve the separation of different structures then additional techniques would be needed; the aim of this study is to showcase the applications of this technique on a small number of model proteins to understand restructuring occurring at low temperatures.

3. *The authors make it a point to quantify changes in the peak width with respect to diffusion, temperature dependent mobility effects, and contributions from conformation changes.*

These are key points and the authors are encouraged to provide additional detail on how these distinctions are made.

The peak width at the zero-field limit can be given from the solution of a diffusing ion plume, namely

$$f(x, t) = g(x, t) * \phi(x, t)$$

Where $f(x, t)$ is the observed ion density, $g(x, t)$ the shape of the injected ion pulse and $\phi(x, t)$ is the shape of the ion swarm starting from a pin-point initial condition (a delta function initial condition), *i.e.* it is the Green's function of the diffusion-advection equation. For a large drift tube this can be given as diffusion in free space, which in one dimension is

$$\phi(x, t) = \frac{A}{(4\pi Dt)^{1/2}} e^{-\frac{(x-vt)^2}{4Dt}}$$

Where A is the total ion density, x the drift length, t the drift time, v the drift velocity and D the longitudinal diffusion coefficient. At the low field limit, D and v are related through the Einstein relation:

$$v = KE = \frac{qE}{k_B T} D$$

which offers a limit on the expected width due to diffusion alone (K being the ion mobility and q the charge of the ion). Now, this situation is idealised since the width also contains the effect of broadening of the peak outside the drift tube, but these are assumed to be small due to the fact that occasionally such narrow peaks are detected experimentally and the time ions spend in the drift tube is greater than any other section between the gating at the beginning and the ToF. But when significantly broader peaks are observed then these hint to additional effects that cannot be attributed to diffusion.

The effects due to temperature on a model "rigid" system is monitored experimentally by looking at the G5 dendrimer ions. It provides a benchmark for the effects due to the increased thermal motion of the collisions and the effects of temperature on the momentum transfer cross-section Ω . Also *cf.* our reply to Reviewer 2, points 1-3.

Peaks that are separated in arrival time but contain ion density between them, as well as very long "tails" in the ATDs can suggest the presence of dynamic transitions, *i.e.* reactions in the drift tube and these are treated as described in our manuscript and SI. Pages 16-18.

4. *Other overarching concerns that the authors are encouraged to recognize, and dismiss if warranted, include whether the observed species across charges states could be derived either through charge reduction during the drift cell or larger oligomers.*

Further, because the temperature is lowered are there any scenarios where solvent adducts exists within the drift cell but are removed prior to mass analysis that might stabilize select conformations?

To check this we have performed aIMS experiments wherein we m/z select each ion prior to activation and in doing this we do not observe any charge-reduced ions at low collision voltages, we have previously demonstrated this for ubiquitin² and G5²³ and have now included data in the SI showing the mass spectra following activation for m/z selected α -synuclein and β -casein (Supplementary information Figure S6; Figure S8) . We infer from this that charge stripping does not contribute significantly to the low temperature cases that we examine with linear-field IMS, further there is no argon or nitrogen in our home-built instrument post drift cell (we do not fill the collision

cell) unlike in Synapts or other commercial IM-MS devices which also would reduce opportunity to charge strip.

We do not observe any significant intensity from water-protein ion clusters in our measurements at any temperature (Supplementary information Figure S 10). We have previously examined the effect of salt adductation as a function of lowered temperature, although for structured proteins and found it can be higher at lower temperatures for some proteins, but at much higher concentrations (2mM NaI) than here¹⁹. In that work, which inspired what we do here we saw only a marginal effect of salt adducts on the CCS of proteins. In general, the lack of adductation on α -synuclein and casein shown here is what we have observed for most IDPs. We are basing our analysis on selecting ions, post acquisition, that are solvent-free and salt-free when detected.

We have added this text on page 13-14 and the corresponding data in the SI.

5. *The use of the dendrimer as a control is a clever approach that helps bound the scope of interpretation. One aspect of the G5 dendrimer is the observation that the peak width actually increases as temperature decreases. This is not what one might expect. Can the authors please address this behavior?*

From the Einstein relation $D = \frac{k_B T}{q} K$, we expect peak widths due to diffusion to increase with temperature. However, at low temperatures the G5 ion appear to have broader ATDs. We interpret this as an effect of ergodicity. Assuming that the conformation energy landscape of the polymer is some energy well, at high temperatures ions have enough time to explore all available structures (which likely correspond to torsional motion of the outermost functional groups). At low temperatures, we still sample all closely related structures, however each individual ion does not have enough time to explore every similar geometry in the millisecond timescale of our experiment. This transition from a non-ergodic ensemble to an ergodic ensemble may manifest itself as a narrowing of the peak width.

We have added this explanation to the main text.

6. *For aSyn data, there is a considerable degree of conformational variation explored by the protein, even across charge states. In fact, there are some distributions (e.g., +7 @250K) that are quite narrow. How do these peaks compare to the theoretical minimum predicted by diffusion and how do these compare to G5? The point of this question is frame the range of different conformations that exist within the larger distributions.*

The resolving power of the IMMS device has been described before²⁴ and is only somewhat less than the theoretical maximum (i.e. purely diffusion limited). The narrowest peaks are slightly wider than what is predicted from diffusion. We discuss this in the manuscript (page 13) and in the supporting data we include the Rmax data for the G5 dendrimer (Supplementary information Table S5). We have added the Rmax data for α -synuclein in the highest (+13) and lowest (+7) charge states to Supplementary information Table S6.

7. *Across the data presented (including the bar chart) the data for 275K shows a considerable degree of conformational flexibility for the protein charge states probed. It is unclear exactly the order of operations for the experimental analysis campaign but is there something special about 275K, was there an experimental differentiator at that temperature, or is this observation simply coincidence?*

This is a main hypothesis that results from our experimental study as well as previous experiments at low temperatures. We detect a structural rearrangement for many protein systems at temperatures below 280K. These rearrangements seem not to be present in As examples of low temperature measurements amass we may one day ascertain if this is a universal property or proteins or just a feature of the proteins we have studied so far.

Many more proteins will have to be studied by this technique, which unfortunately is not yet a streamlined high throughput technique, to determine if such a behaviour could be universal. But the experimental evidence so far suggests that it might be.

In the current study we also note that the conformational heterogeneity during this transition is more pronounced for IDPs than it is for globular proteins, and absent for the G5 dendrimer.

8. *While asking for additional data is somewhat contentious, the work by Poyer et al. probing gas phase isomerism provides an interesting framework to quantify changes from a thermodynamic perspective. Is there any chance some of the species probed are in some sort of thermal equilibrium that can be captured on the experimental timescale of the ion flight time or are the species observed reasonably static? Is there a way to thermodynamically quantify transitions between temperatures?*

We do believe that the $[M+13H]^{13+}$ ion of α -synuclein does undergo a structural transition that is detected. Further possibilities may exist but without being certain of the parent species (which may originate from a different m/z). That result has been analysed in our study.

Peak width can hint as well to a conformational equilibrium being reached. As we mentioned at point (5) above, this transition from a non-ergodic ensemble to an ergodic ensemble may manifest itself as a narrowing of the peak width. However, in the absence of more certain experimental evidence either from complementary techniques or by choosing different rigid molecules, combined with extensive simulations, this interpretation is still too tentative to be published at this stage.

Below are specific comments related to the text:

Abstract:

1. L18 — *How do you “provide the role of temperature?”*
We have changed this sentence as ‘measure the effect of temperature on conformational landscapes’
2. L19 — *There are a variety of collision effects. How do you differentiate between the range of types (e.g., collisional cooling/heating, dissociative, etc.)?*
3. L23 — *Should read: “does not alter significantly over this temperature range.”*
Done

Introduction

4. *The authors should make it clear that the Mason Schamp equation is a first approximation at best. Including the concept of a momentum-transfer cross-section would also help bound expectations for the veracity of this equation.*

We have changed the introduction and mention explicitly the temperature effects which are included in Ω as the reviewer suggests.

5. L42 — *ExcellIMS does sell variable temperature drift cell but it is aimed squarely at the high temperature range.*

We have specifically defined the temperature range to below 295K and note this supplier of higher temperature IMS devices.

6. L82 — *Should read: “requirements of such an instrument,”*

We have replaced sentences with more accurate expressions.

7. L84 — *A reference should be included on this line.*

Done.

8. L136 — *Why are the proteins examined “exemplar?”*

The definition of exemplar is ‘a person or thing serving as a typical example or appropriate model’.

We have chosen proteins that have been intensively studied, albeit not in the low pressure and low temperature regime. Ubiquitin has been studied broadly, including by IMMS. As a protein with a small molecular weight and a highly conserved structure, the variable temperature IMS data of its native form has also been well described. Here, Ubiquitin is sprayed from methanol-water solution as a comparison to compare it to β -casein and α -synuclein. β -casein and α -synuclein are typical IDPs, β -casein as a milk component is an important protein in food and α -synuclein is implicated in Parkinson’s disease. They are widely available proteins that are used as “model” IDPs and for that reason we deem they are well suited for showcasing the technique and to observe the behaviour or conformationally heterogeneous proteins. Our study does show that the behaviour of those IDPs is different than that of polymers.

Instrumentation

9. *Are the authors confident that the ions reach thermal equilibrium prior to injection into the mobility cell? The “ion buncher” is quite short in the manuscript referenced by the authors. Essentially, making a statement related to concept that the ion are thermalized to the experimental temperature prior to injection would be helpful.*

Ions spend up to 28ms in the ion trap prior to being released into the drift region, which is longer than the drift time of the ions. This stage happens before the beginning of the ion mobility experiment. Given the pressure (c. 2.2 Torr) in the region and the complete transmission with the drift region the residence time alone allows us to infer that ions are thermalised before starting the ion mobility experiment. The ion buncher operates at pressures and temperatures equal to that of the drift tube. The length of the ion buncher (5cm) ensures that ions can thermalise at this length, which is similar to earlier designs of lower-resolution drift tubes. At the pressures operated, it has been shown that very short lengths are sufficient to ensure thermalisation of the ions²⁵.

10. L230 — *Can the authors use a different word than “beautifully?”*

We have rephrased that sentence.

Reference

- (1) Norgate, E. L.; Upton, R.; Hansen, K.; Bellina, B.; Brookes, C.; Politis, A.; Barran, P. E. Cold Denaturation of Proteins in the Absence of Solvent: Implications for Protein Storage**. *Angew. Chem. - Int. Ed.* **2022**. <https://doi.org/10.1002/anie.202115047>.
- (2) Ujma, J.; Jhingree, J.; Norgate, E.; Upton, R.; Wang, X.; Benoit, F.; Bellina, B.; Barran, P. Protein Unfolding in Freeze Frames: Intermediate States Are Revealed by Variable-Temperature Ion Mobility-Mass Spectrometry. *Anal. Chem.* **2022**. <https://doi.org/10.1021/acs.analchem.2c03066>.
- (3) Marcoux, J.; Robinson, C. V. Twenty Years of Gas Phase Structural Biology. *Structure* **2013**, *21* (9), 1541–1550. <https://doi.org/10.1016/j.str.2013.08.002>.
- (4) Topf, M.; Lasker, K.; Webb, B.; Wolfson, H.; Chiu, W.; Sali, A. Protein Structure Fitting and Refinement Guided by Cryo-EM Density. *Structure* **2008**, *16* (2), 295–307. <https://doi.org/10.1016/j.str.2007.11.016>.
- (5) Hänsel, R.; Luh, L. M.; Corbeski, I.; Trantirek, L.; Dötsch, V. In-Cell NMR and EPR Spectroscopy of Biomacromolecules. *Angew. Chem. Int. Ed.* **2014**, *53* (39), 10300–10314. <https://doi.org/10.1002/anie.201311320>.
- (6) Cvetkovska, M.; Hüner, N. P. A.; Smith, D. R. Chilling out: The Evolution and Diversification of Psychrophilic Algae with a Focus on Chlamydomonadales. *Polar Biol.* **2017**, *40* (6), 1169–1184. <https://doi.org/10.1007/s00300-016-2045-4>.
- (7) Collins, T.; Margesin, R. Psychrophilic Lifestyles: Mechanisms of Adaptation and Biotechnological Tools. *Appl. Microbiol. Biotechnol.* **2019**, *103* (7), 2857–2871. <https://doi.org/10.1007/s00253-019-09659-5>.
- (8) Levental, I.; Lyman, E. Regulation of Membrane Protein Structure and Function by Their Lipid Nano-Environment. *Nat. Rev. Mol. Cell Biol.* **2023**, *24* (2), 107–122. <https://doi.org/10.1038/s41580-022-00524-4>.
- (9) Contreras, F.-X.; Ernst, A. M.; Wieland, F.; Brügger, B. Specificity of Intramembrane Protein–Lipid Interactions. *Cold Spring Harb. Perspect. Biol.* **2011**, *3* (6), a004705. <https://doi.org/10.1101/cshperspect.a004705>.
- (10) Zhu, H.; Yang, F.; Zhu, Y.; Li, A.; He, W.; Huang, J.; Li, G. Investigation of Dielectric Constants of Water in a Nano-Confined Pore. *RSC Adv.* **2020**, *10* (15), 8628–8635. <https://doi.org/10.1039/C9RA09399K>.
- (11) Allison, T. M.; Reading, E.; Liko, I.; Baldwin, A. J.; Laganowsky, A.; Robinson, C. V. Quantifying the Stabilizing Effects of Protein–Ligand Interactions in the Gas Phase. *Nat. Commun.* **2015**, *6* (1), 8551. <https://doi.org/10.1038/ncomms9551>.
- (12) Beveridge, R.; Migas, L. G.; Payne, K. A. P.; Scrutton, N. S.; Leys, D.; Barran, P. E. Mass Spectrometry Locates Local and Allosteric Conformational Changes That Occur on Cofactor Binding. *Nat. Commun.* **2016**, *7*. <https://doi.org/10.1038/ncomms12163>.
- (13) Jurneczko, E.; Barran, P. E. How Useful Is Ion Mobility Mass Spectrometry for Structural Biology? The Relationship between Protein Crystal Structures and Their Collision Cross Sections in the Gas Phase. *Analyst* **2010**, *136* (1), 20–28. <https://doi.org/10.1039/C0AN00373E>.
- (14) Westphall, M. S.; Lee, K. W.; Salome, A. Z.; Lodge, J.; Grant, T.; Coon, J. J. 3D Structure Determination of Protein Complexes Using Matrix-Landing Mass Spectrometry. *bioRxiv* **2021**, 2021.10.13.464253. <https://doi.org/10.1101/2021.10.13.464253>.
- (15) Ouyang, Z.; Takáts, Z.; Blake, T. A.; Gologan, B.; Guymon, A. J.; Wiseman, J. M.; Oliver, J. C.; Davisson, V. J.; Cooks, R. G. Preparing Protein Microarrays by Soft-Landing of Mass-Selected Ions. *Science* **2003**, *301* (5638), 1351–1354. <https://doi.org/10.1126/science.1088776>.
- (16) El-Baba, T. J.; Woodall, D. W.; Raab, S. A.; Fuller, D. R.; Laganowsky, A.; Russell, D. H.; Clemmer, D. E. Melting Proteins: Evidence for Multiple Stable Structures upon Thermal Denaturation of Native Ubiquitin from Ion Mobility Spectrometry-Mass Spectrometry Measurements. *J. Am. Chem. Soc.* **2017**, *139* (18), 6306–6309. <https://doi.org/10.1021/jacs.7b02774>.

- (17) Dobo, A.; Kaltashov, I. A. Detection of Multiple Protein Conformational Ensembles in Solution via Deconvolution of Charge-State Distributions in ESI MS. *Anal. Chem.* **2001**, *73* (20), 4763–4773. <https://doi.org/10.1021/ac010713f>.
- (18) Cole, H.; Porrini, M.; Morris, R.; Smith, T.; Kalapothakis, J.; Weidt, S.; Mackay, C. L.; MacPhee, C. E.; Barran, P. E. Early Stages of Insulin Fibrillogenesis Examined with Ion Mobility Mass Spectrometry and Molecular Modelling. *Analyst* **2015**, *140* (20). <https://doi.org/10.1039/c5an01253h>.
- (19) Berezovskaya, Y.; Porrini, M.; Barran, P. E. P. E. The Effect of Salt on the Conformations of Three Model Proteins Is Revealed by Variable Temperature Ion Mobility Mass Spectrometry. *Int. J. Mass Spectrom.* **2013**, *345–347*, 8–18. <https://doi.org/10.1016/j.ijms.2013.02.005>.
- (20) Merenbloom, S. I.; Flick, T. G.; Daly, M. P.; Williams, E. R. Effects of Select Anions from the Hofmeister Series on the Gas-Phase Conformations of Protein Ions Measured with Traveling-Wave Ion Mobility Spectrometry/Mass Spectrometry. *J. Am. Soc. Mass Spectrom.* **2011**, *22* (11), 1978–1990. <https://doi.org/10.1007/s13361-011-0238-1>.
- (21) Black, R.; Barkhanskiy, A.; Ramakers, L. A. I.; Theisen, A.; Brown, J. M.; Bellina, B.; Trivedi, D. K.; Barran, P. E. Characterization of Native Protein Structure with Ion Mobility Mass Spectrometry, Multiplexed Fragmentation Strategies and Multivariant Analysis. *Int. J. Mass Spectrom.* **2021**, *464*, 116588. <https://doi.org/10.1016/j.ijms.2021.116588>.
- (22) Breuker, K.; Brüscheiler, S.; Tollinger, M. Electrostatic Stabilization of a Native Protein Structure in the Gas Phase. *Angew. Chem. - Int. Ed.* **2011**, *50* (4), 873–877. <https://doi.org/10.1002/anie.201005112>.
- (23) Benoit, F.; Wang, X.; Dai, J.; Geue, N.; England, R. M.; Bristow, A. W. T.; Barran, P. E. Exploring the Conformational Landscape of Poly(L-Lysine) Dendrimers Using Ion Mobility Mass Spectrometry. *Anal. Chem.* **2024**, *96* (23), 9390–9398. <https://doi.org/10.1021/acs.analchem.4c00099>.
- (24) Ujma, J.; Giles, K.; Morris, M.; Barran, P. E.; Spectrometry, C. M. New High Resolution Ion Mobility Mass Spectrometer Capable of Measurements of Collision Cross Sections from 150 to 520 K. *Anal. Chem.* **2016**, *88* (19), 9469–9478. <https://doi.org/10.1021/acs.analchem.6b01812>.
- (25) Wyttenbach, T.; Paizs, B.; Barran, P.; Breci, L.; Liu, D.; Suhai, S.; Wysocki, V. H.; Bowers, M. T. The Effect of the Initial Water of Hydration on the Energetics, Structures, and H/D Exchange Mechanism of a Family of Pentapeptides: An Experimental and Theoretical Study. *J. Am. Chem. Soc.* **2003**, *125* (45), 13768–13775.

REVIEWER COMMENTS

We thank all the reviewers for their suggestions that have led to improving our manuscript.

Reviewer #1 (Remarks to the Author):

The authors have addressed the concerns of my initial critique. I recommend for publication.

We are grateful that the reviewer has approved our manuscript and are thankful for their suggestions.

Reviewer #2 (Remarks to the Author):

The authors have significantly improved the manuscript with this revised version. I only have a few minor points that they might want to consider.

We are very grateful for the reviewer's suggestions and have made the relevant edits.

The revised text in the Introduction does a better job of explaining the two different effects of temperature on measured CCS values assuming no structural changes. The claim (made in the Introduction and repeated on line 227) that CCS is expected to scale with the inverse square root of temperature might still not be obvious to the non-specialist reader though, as it relies on the assumption that the product of these two values remains constant. As the authors already stated in their reply, the validity of this assumption depends on how the momentum-transfer integral changes as a function of temperature, and on whether the long-range polarisation potential is the dominant factor for interactions. This seems realistic under the low-temperature conditions used in this work, and the text alludes to this assumption by stating that long-range attractive forces dominate at low temperatures. However, I believe it would be useful to add one or two sentences to emphasise this point in a bit more detail in order to avoid potential confusion (perhaps even introducing the concept of a momentum-transfer cross section as was suggested by Reviewer 3) as the text near the end of page 3, and the positioning of Equation 1, currently might give the impression that this correlation can be derived from the Mason-Schamp equation.

We have made an additional clarification of this point in lines 60 and 65-69 of the new manuscript. We kept the phrasing of line 227 in light of those additions in the introduction.

That said, this is a fairly minor point that does not detract from the significance of the work, and I fully agree with the authors that temperature-dependent structural changes are the most interesting effect studied in this work, and that there is a clear difference between the behaviour of systems where such structural changes are limited (such as the dendrimer) and those where they are more pronounced.

We thank the reviewer for noting this finding and it also supports our decision to retain the CCS axis range shown in Figure 3 (see below).

In Figure 3, very broad CCS ranges with a lot of white space are shown, in particular for the dendrimer, ubiquitin, and beta-casein. This makes it difficult to see details of the distribution as well as small CCS differences between measurement temperatures. Providing links to interactive plots for the reader to explore is nice, but perhaps the distributions in the figure could be zoomed in a bit.

We appreciate this suggestion; we have opted to use the same CCS range for protein ions in figures 3, 4 and 5, which is important for readers make a direct visual comparison between all those ATDs and our interactive plots allow zoom.

Minor points:

On line 49, the fragment 'giving rise to the observed a greater CCS values' has grammar issues. On the line before that, 'the collision trajectory of the ion-neutral pair is farm more influenced' has a typo.

On line 280, the authors mention 'sharp conformers.' The signals associated with the conformers are sharp, not the conformers themselves.

We have corrected these mistakes.

Reviewer #3 (Remarks to the Author):

Based upon the response to the reviewers, there appears to be some concern related to the novelty approach and the relevance of the gas phase approaches. While the point of Reviewer #1 with respect to water is noted, the points made by the authors are salient. While it is true that the impact of bulk water is largely removed, an extension of Reviewer #1's concern to other techniques would also preclude the rigorous interpretation of biological thermodynamics using those approaches--a crystal structure, by definition, is derived from non-native conditions. Overall, the authors have provided detailed responses to most concerns, emphasizing the novelty of their findings, refining the clarity of their explanations, and making necessary revisions to improve the manuscript.

We thank the reviewer for highlighting the unique advantages of our approach and their appreciation of the modifications we have made based on the combined comments from the reviewers, and how they improve the clarity of our findings.

There are a few points worth noting that could further aid in developing the manuscript. Aside from the estimation of CCS, perhaps the most quantitative work presented is the data from the Arrhenius plot in the Supp Material. The authors state

that the parameters for this plot are derived from a fit of Equation 2 in the Supp Material to the experimental data. Moreover, they noted that, "**The blue solid line presents a good fit to the experimental ATD.**" However, these fits are not shown. Are they suggesting that the light blue line shown in Figure 6 of the main manuscript are those fits? If so, this is not clear in the manuscript.

We have clarified in the SI that there are references to Figure 6b of the main text. We have edited the legends of the main text Figure 6b to state that the solid traces are fits to the experimental data.

Finally, given the need to clearly define the methods and approach, perhaps the authors should simply show these fits that go into the Arrhenius plot and release the CSVs and associated code contributing to those fits. This would further support the method they aim to expand upon and alleviate some of the concerns raised by Reviewer #1.

We have supplied the fit of the data using Equations [2] and [3] of the SI in .csv format as well as the Origin workbook that is used to analyse the rate constant from our data. We have supplied these for review and added them to the figshare data repository (<https://doi.org/10.6084/m9.figshare.25145867>) for this work which will be made public on paper acceptance.

With minor clarifications, this manuscript warrants publication as there are few experimental approaches that can provide detailed insights that complement the more common interpretations derived from the canonical averages derived from NMR, crystal structures, and microscopy techniques.

We thank the reviewer for their approval.

Minor Comments:

Spacing between reported values and units should be included (e.g., 273 K and not 273K).

m/z is italicized.

Typo in the supplemental material: "which equal to $1-t_A/t_B$, β and θ "

We have corrected these errors.